# Learn to Merge: Meta-Learning for Adaptive Multi-Task Model Merging

## Abstract

Model merging in the pretrain-finetune paradigm has proven effective by combining multiple finetuned models into one with multi-task capabilities. Recent merging methods aim to boost merged models' performance through strategies such as mitigating conflicts, adding trainable modules, and incorporating task-specific components. In most methods, the parameter merging procedure is based on Task Arithmetic, a widely used technique that creates task vectors from each finetuned model and linearly combines them with coefficients into consolidated model parameters. Except for studies specifically focusing on the merging coefficients, many other methods treat them as hand-tuned hyperparameter. However, the merging coefficients, which govern the entire merging process including the subsequent module training, are empirically crucial for achieving optimal performance and tradeoff across tasks. Thus, this paper proposed an innovative model merging framework called MetaMerging, which constructs the merged model with unified model and learnable lightweight task-specific adapters. Specifically, we introduce a novel meta-learning algorithm to adaptively optimize the merging coefficients for computing the unified model, which enhances its generalization and enables more effective adapter training. Extensive experiments on CV and NLP fields show strong performance of MetaMerging on various downstream tasks and demonstrate the effectiveness of meta-learning in our method compared to other parameter merging methods. Our code is available at https://anonymous.4open.science/r/MetaMerging-53A1

## 1 Introduction

With the rapid development of deep learning, model architectures are scaling up and the computational burden of training is increasing (Krizhevsky et al., 2009; Dosovitskiy et al., 2020; Vaswani et al., 2017; He et al., 2016). Consequently, the pretrain-finetune paradigm has become a widely adopted approach, where a foundation model is first pretrained and then finetuned for multiple downstream tasks (Liu et al., 2024; Wortsman et al., 2022b; Li et al., 2025; He et al., 2022). This usually results in multiple finetuned models, while maintaining separate models for multiple tasks results in substantial storage and computational costs. To address this, model merging has been proposed as an effective method for constructing a unified multi-task model by leveraging the knowledge of multiple finetuned models, and it has quickly become a popular research direction (Yang et al., 2024a). This paradigm of "learning from models" (Zheng et al., 2025) is viewed as a strong complement to the traditional "learning from data," as merged models can effectively consolidate the knowledge inherent in fine-tuned models while enhancing overall parameter efficiency.

Early model merging methods (such as Weight Averaging (Wortsman et al., 2022a), Task Arithmetic (Ilharco et al., 2023), Ties-Merging (Yadav et al., 2023)) mainly focused on parameter-level operations, which are simple and efficient but often lead to a prominent performance gap between the merged model and the individually finetuned models. Such techniques are built on the task vector, which serves as a unique representation for a particular task and can be simply represented as $V_k = \theta_{ft\_k} - \theta_{pre}$, where $\theta_{ft\_k}$, $\theta_{pre}$ denote the parameters of finetuned model of task $k$ and pretrained model, respectively. The unified model is then computed by linearly combining these task vectors with merging coefficients $\{\lambda_k\}_{k=1}^K$, as equation in Figure 2. More recently, advanced merging methods (such as Surgery (Yang et al., 2024b), Twin-Merging Lu et al., WEMoE (Shen et al., 2024)) have introduced lightweight unsupervised training and additional components (e.g., task-specific

weights, router modules) to enhance the merged model and improve performance. In general, these methods follow two sequential steps: ❶ merging the parameters of finetuned models into a unified model through conventional merging methods; ❷ training additional adapter modules based on the unified model. For example, in the Surgery method, the final multi-task model is constructed from two components: the unified model and task-specific adapters. This design separates shared knowledge from task-specific adaptations, improving scalability and parameter efficiency.

However, these methods emphasize the subsequent training phase but naively treat the merging coefficient $\{\lambda_k\}_{k=1}^K$ as manually tuned hyper-parameters or a uniform value $1/K$ (i.e. Weight Averaging). As shown in Figure 1, under the same adaptation process, unified models produced by different merging strategies benefit unevenly from adapters, leading to substantial discrepancies in final performance. This is because the existing progressive knowledge modularization strategy faces significant limitations. First, the quality of the unified model heavily depends on the initial merging methods. If the merging process fails to properly balance the representations of each task, the unified model may favor certain tasks while neglecting others. In such cases, adding additional adapter modules may not fully

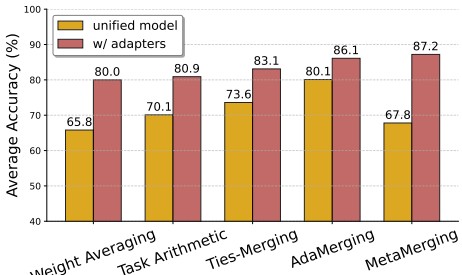

Figure 1: Multi-task performance of unified models from different merging methods on eight vision tasks, evaluated before and after task-specific adapter training.

compensate for the representation deficiencies of the unified model in certain tasks. Second, these methods lack the ability to consider the training needs of downstream modules during the merging process, leading to a disconnect between the merging results and task adaptations, and failing to form effective synergy. Thus, a key research question arises: *How can we learn a strong initialization for the unified model, tailored to the training needs of downstream adapters, thereby achieving more balanced and effective multi-task performance?*

Motivated by these observations, we introduce a meta-learning algorithm and adaptively learn better merging coefficients to obtain a more generalizable and task-balanced unified model, which can better support the training of subsequent task-specific adapters for the given tasks, as illustrated in Figure 2. The inspiration comes from MAML (Finn et al., 2017), a meta-learning algorithm that learns a good model initialization such that it can be quickly adapted to new tasks with only a few gradient updates. In this paper, we propose a novel model merging framework, MetaMerging, which constructs the merged model by integrating a unified model with task-specific adapters. In the inner loop, we perform rapid provisional gradient updates to imitate adapter training, and in the outer loop, we meta-update the merging coefficients to im-

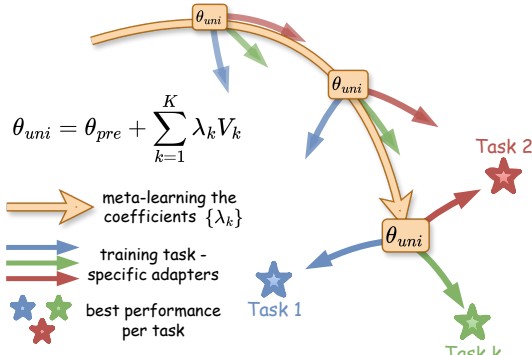

Figure 2: Diagram of our meta-learning process. The merging coefficients are optimized to produce a more generalized unified model, which enables more effective task-specific adapter training.

prove inference performance by simulating test-time adapter usage, as shown in Figure 3. In this way, we can learn the best way to merge task vectors and automatically find the most generalized unified model, thereby enhancing the final performance after incorporating adapters. Interestingly, as the results of MetaMerging shown in Figure 1, we found the optimal unified model is not necessarily the one that performs best on all tasks, but the one that better preserves the potential for adapters, thereby enabling greater downstream gains.

To summarize, our contributions are as follows: (1) We propose MetaMerging, an innovative model merging framework that substantially improves the multi-task performance of the merged model and is effective across a wide range of tasks. (2) We develop a meta-learning algorithm that autonomously learns to merge task vectors with proper coefficients, introducing a new direction for designing more effective and sophisticated model merging methods. (3) We conduct extensive experiments on various tasks and models, verifying the effectiveness of MetaMerging and enhancing its interpretability.

## 2 RELATED WORK

**Model Merging.** Recently, model merging has gained attention as a viable alternative to conventional multi-task and transfer learning paradigms (Yang et al., 2024a; Wan et al., 2024; Tang et al., 2024b; Xiong et al., 2024; Zhou et al., 2024a). Instead of jointly training multiple tasks or relying on continual learning schemes, model merging directly constructed a unified model using the existing parameters of task-specific finetuned models. A straightforward strategy for model merging is to average the model weights (Wortsman et al., 2022a), but this often leads to substantial performance degradation. Early methods such as Task Arithmetic and Ties-Merging mainly operate in the parameter space, for example by adjusting task-vector coefficients or applying sparsification, in order to alleviate task conflicts and improve the merged model's performance and generalization (Ilharco et al., 2023; Yadav et al., 2023; Yang et al., 2024d; Yu et al., 2024). Subsequent works have introduced more sophisticated merging strategies, such as task-specific module, optimal transport in parameter space, and router-based method, aiming to further narrow the performance gap between merged and finetuned models (Chen and Kwok, 2024; Lu et al.; Shen et al., 2024; Huang et al., 2024; Yang et al., 2024c). They typically follow the merging-and-training paradigm introduced in Section 1, which requires only the input data but not the labels of downstream tasks. Building upon such advanced structures, our method leverages meta-learning to produce a more generalizable unified model, which effectively coordinates parameter merging with subsequent training to achieve superior performance.

**Meta-Learning.** Meta-learning, also called "learning to learn", provides a general framework to acquire transferable knowledge across tasks (Han et al., 2021; Khoee et al., 2024; Vettoruzzo et al., 2024; Finn et al., 2018). The essence of meta-learning is that, rather than addressing tasks independently with a pre-defined algorithm, it improves the learning procedure by exploiting knowledge gained over repeated learning episodes (Hospedales et al., 2021). Classical approaches, such as meta-metric, meta-optimization, and model-based meta-learning, have been widely applied in few-shot and multi-task settings (Yuan et al., 2020). Among them, A classical method, MAML (Finn et al., 2017), aim to train initialization parameters that can be rapidly adapted to new tasks with limited data, while task-specific learners are guided by meta-gradients. Recent advances have extended meta-learning to parameter-efficient adaptation, neural architecture search, and multi-modal scenarios (Zhou et al., 2024b; Wang et al., 2022). In this paper, meta-learning has been used to learn the merging coefficients when combining task vectors, enabling more adaptive and generalized unified model and more effective subsequent adapter training.

## 3 METHODOLOGY

In this section, we introduce MetaMerging, our framework for merging multiple task-specific models into a single, efficient multi-task model. Starting from a pretrained backbone and its fine-tuned variants. Specifically, we first represent each task by a task vector—the difference between its fine-tuned model and the pretrained model, and combine these vectors into a unified model using a set of merging coefficients. We then attach lightweight adapters for each task on top of this unified model. The key idea is to meta-learn the merging coefficients so that, after a few steps of adapter training on unlabeled data, the unified model plus adapters mimic the original fine-tuned models as well as possible. In Section 3.2, we present the overall framework of our method and describe how the target multi-task model is constructed. Finally, in Section 3.3, we elaborate on our key contribution: the concrete meta-learning algorithm for adaptively finding the optimal merging coefficients of task vectors. The overall workflow of MetaMerging is summarized in Algorithm 1.

### 3.1 PRELIMINARIES

Let $f_\theta(\mathbf{x}_i) \to \mathbf{h}_i$ be a neural network encoder model parameterized by $\theta$, where $\mathbf{x}_i \in \mathbb{R}^n$ is the input data and the $\mathbf{h}_i \in \mathbb{R}^d$ is the final feature with dimension $d$ extracted by $f_\theta$. Let $g_\phi(\mathbf{h}) \to \hat{\mathbf{y}}_i$ be a classification head parameterized by $\phi$, producing the prediction logits $\hat{\mathbf{y}}_i \in \mathbb{R}^c$, where $c$ is the number of classes for the final classification. So for a specific downstream task $k$, the inference process can be formulated as: $\hat{\mathbf{y}}_i = g_{\phi_k}(f_{\theta_k}(\mathbf{x}_i))$. In the pretrain-finetune paradigm, a pretrained model $\theta_{pre}$, generally endowed with fundamental knowledge of a domain such as language or images, is fine-tuned on multiple downstream tasks with their respective datasets $\mathcal{D}_k$ to get task-specific models $\theta_{ft\_k}$ that achieve improved performance. Now there are $K$ tasks with datasets $\{\mathcal{D}_1, \mathcal{D}_2, ..., \mathcal{D}_K\}$,

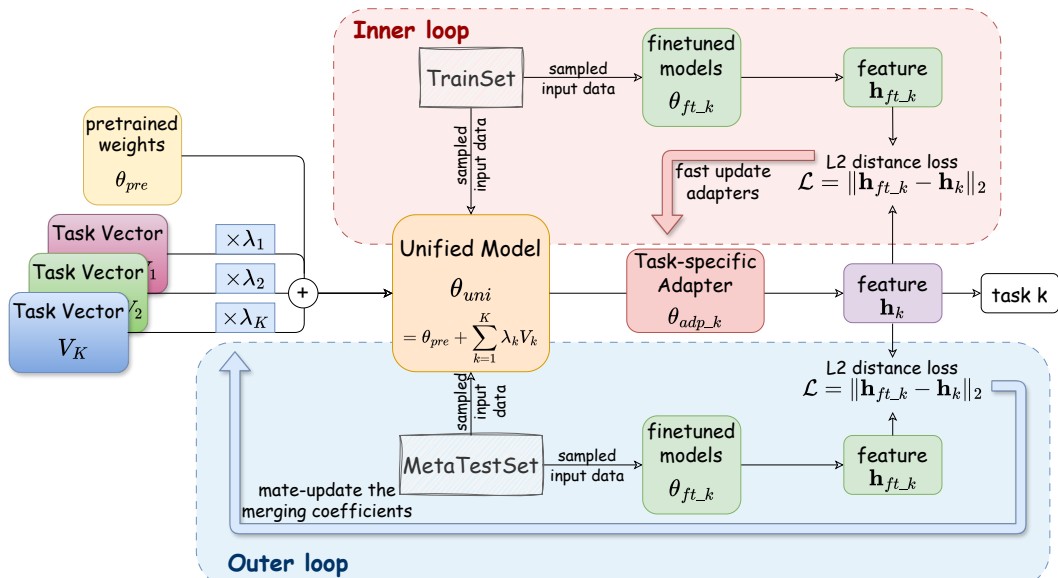

Figure 3: The overall framework of MetaMerging consists of two stages: (1) constructing the unified model and (2) training task-specific adapters. The unified model is obtained by merging pretrained parameters and task vectors with coefficients, which are learned via our meta-learning algorithm. In each meta cycle, we first perform a fast update of the adapters and then update the coefficients $\{\lambda_k\}_{k=1}^K$ using the meta-gradient. In this way, we leverage episodic adapter training to optimize a better unified model, thereby improving the effectiveness of subsequent adapter training.

and $K$ models $\{\theta_1, \theta_2, ..., \theta_K\}$, each equipped with task specific classification head $\{\phi_1, \phi_2, ..., \phi_K\}$, finetuned from same pretrained model $\theta_{pre}$. The goal of model merging is efficiently obtain a unified model $\theta_{uni}$ from existing $\{\theta_1, \theta_2, ..., \theta_K\}$ and $\theta_{pre}$ without using any labeled data. After getting the merged model $\theta_{uni}$, multi-task inference can be performed as: $\hat{\mathbf{y}}_i = g_{\phi_k}(f_{\theta_{uni}}(\mathbf{x}_i))$, which significantly reduces the total number of parameters, alleviating both storage and computational costs. The classification head is excluded from the merging procedure, as the number of classes varies across tasks.

With the development of model merging research, recent methods extend beyond simple parameter-level merging, aiming to further improve multi-task performance. In addition to the unified model $\theta_{uni}$, they incorporate extra lightweight modules trained on unlabeled input data from $\mathcal{D}_k$, which can be viewed as a form of test-time adaptation. Notably, the training time is generally short, and the total number of parameters remains similar to that of a single unified model, making their method remain competitive with multi-task learning. In conclusion, a model merging method should: (1) Maintain efficient storage and inference, much smaller than deploying all fine-tuned models. (2) Require only lightweight additional training compared to joint multi-task learning from scratch. (3) Avoid reliance on labeled data from downstream tasks.

## 3.2 OVERALL FRAMEWORK

Building on insights from previous studies (Lu et al.; Shen et al., 2024; Yang et al., 2024b), our method decomposes knowledge into shared knowledge and task-specific knowledge, and constructs the final merged model using a unified model and lightweight task-specific adapters, as illustrated in Figure 3. The method consists of two stages: (1) merging task vectors to obtain the unified model, and (2) training adapters on the fixed unified model.

The unified model represents the shared knowledge across the $K$ tasks, serving as a compressed abstraction of the task-specific expertise from different fine-tuned models. The features extracted by the unified model can be generally applicable to multiple tasks. In our method, we compute the unified model by linearly combining the task vectors, a technique widely adopted in prior works. This can be formulated as: $\theta_{uni} = \theta_{pre} + \sum_{k=1}^K \lambda_k V_k$, where $V_k = \theta_{ft\_k} - \theta_{pre}$ denotes the task

---

**Algorithm 1** MetaMerging

---

**Require:** fine-tuned model $\{f_{\theta_1}, f_{\theta_2}, ..., f_{\theta_K}\}$, pretrained model $f_{\theta_{pre}}$, unlabeled datasets $\mathcal{D}_k$, initialized
   task-specific adapters $\{A_{\theta_{ada\_1}}, A_{\theta_{ada\_2}}, ..., A_{\theta_{ada\_K}}\}$, Merging coefficients $\{\lambda_k\}_{k=1}^K$, loss fuction $\mathcal{L}$ (L2
   distance), step size hyperparameters $\alpha, \beta$

1: **Coarsely merging:**
2: Compute the unified model $f_{\theta_{uni}}$:
3:    $\theta_{uni}(\lambda) = \theta_{pre} + \sum_{k=1}^K \lambda_k (\theta_k - \theta_{pre})$

4: **Meta-learning merging coefficients:**
5: **while** not done **do**
6:     **for** each task $k$ **do**
7:         Sample inputs $\mathbf{X}_i$ from train dataset $\mathcal{D}_k$
8:         Compute alignment loss: $\mathcal{L}_k = \|f_{\theta_k}(\mathbf{X}_i) - A_{\theta_{ada\_k}}(f_{\theta_{uni}}(\mathbf{X}_i))\|_2$
9:         Perform a one-step gradient update of the task-specific adapter from its initialization:
10:         $\theta'_{ada\_k} = \theta_{ada\_k} - \alpha \nabla_{\theta_{ada\_k}} \mathcal{L}_k$
11:         Sample test inputs $\mathbf{X}'_i$ from meta-test dataset $\mathcal{D}'_k$ for upper-level mate-update
12:         Compute the test metric as meta loss: $\mathcal{L}'_k = \|f_{\theta_k}(\mathbf{X}'_i) - A_{\theta'_{ada\_k}}(f_{\theta_{uni}}(\mathbf{X}'_i))\|_2$,
13:         imitating the test-time inference of trained adapters on downstream task.
14:     **end for**
15:     Update $\{\lambda_k\}_{k=1}^K$ with gradient descent: $\lambda \leftarrow \lambda - \beta \nabla_\lambda \sum_{k=1}^K \mathcal{L}'_k$
16: **end while**

17: **Training Adapters:**
18: **for** each task $k$ **do**
19:     **while** not convergent **do**
20:         Sample inputs $\mathbf{X}_i$ from train datasets $\mathcal{D}_k$
21:         Train the task-specific adapter $\theta_{ada\_k}$ from initialization,
22:         by minimizing the loss: $\mathcal{L}_k = \|f_{\theta_k}(\mathbf{X}_i) - A_{\theta_{ada\_k}}(f_{\theta_{uni}}(\mathbf{X}_i))\|_2$
23:     **end while**
24: **end for**

---

**Ensure:** Obtain optimal $\theta_{uni}$ with respect to $\{\lambda_k\}_{k=1}^K$ and final trained $\{\theta_{ada\_1}, \theta_{ada\_2}, ..., \theta_{ada\_K}\}$

---

vector of task $k$, and $\lambda_k$ is the merging coefficient controlling the balance across tasks. Unlike other methods that manually tune the merging coefficients $\{\lambda_k\}_{k=1}^K$, we optimize them via meta-learning to obtain a generalizable unified model, which enables more effective task-specific adapters training. This will be discussed in detail in Section 3.3.

The task-specific adapters is added after the feature extracted by unified model, serving as task-specific knowledge to narrow the "representation bias" comparing the finetuned models (Yang et al., 2024c). Without loss of generality, we construct the adapter using two fully connected layers (see Appendix C), and other lightweight implementations are also possible. Inspired by Surgery (Yang et al., 2024b), we train the adapters by minimizing a feature alignment loss. Specifically, we denote the adapter layer as $A_\theta(\cdot)$, and let $\{\theta_{ada\_1}, \theta_{ada\_2}, ..., \theta_{ada\_K}\}$ be the set of parameters for all tasks. For task $k$, given the input $\mathbf{x}$ sampled from $\mathcal{D}_k$, we first perform a forward pass through the finetuned model to obtain $\mathbf{h}_{ft\_k} = f_{\theta_{ft\_k}}(\mathbf{x})$, an then perform a forward pass through the unified model $\theta_{uni}$ followed by the task-specific adapter $\theta_{ada\_k}$ to obtain $\mathbf{h}_k = A_{\theta_{ada\_k}}(f_{\theta_{uni}}(\mathbf{x}))$. Our goal is to mitigate the discrepancy between $\mathbf{h}_k$ and $\mathbf{h}_{ft\_k}$, thereby improving the final performance. Accordingly, the loss function is defined as $\mathcal{L}_k = \|\mathbf{h}_{ft\_k} - \mathbf{h}_k\|_2$, where $\|\cdot\|_2$ denotes the L2 norm. In essence, this process is analogous to knowledge distillation from fine-tuned models. Notably, the training is highly efficient, empirically requiring only one epoch over the validation set compared to 30-50 epochs in standard multi-task learning.

## 3.3 OPTIMIZE THE MERGING COEFFICIENTS VIA META-LEARNING

Here, we describe the merging coefficient optimization strategy used in stage (1) of Section 3.2. Recall the reseach question raised in Section 1, we seek a more generalizable unified model that is easy to adapt to downstream tasks, thus supporting subsequent adapter training. Driven by this, we turn to Meta-learning, which is the process of distilling the experience of multiple learning episodes

and using this experience to improve future learning performance (Hospedales et al., 2021). In this paper, our goal is to improve the effectiveness of adapter learning. Therefore, a natural idea is to generate episodes of adapter learning and leverage their experience to improve the adapter learning process itself. In each episode, we use small batches and few updates to rapidly simulate the process of adapter training and testing. And the meta-updated parameters are the merging coefficients used in computing the unified model, which serve as an influential initialization for adapter training.

Specifically, we first split each task dataset into a train set $\mathcal{D}_k$ and a test set $\mathcal{D}'_k$. Within a meta-update cycle, for each task $k$, we sample a batch $\mathbf{X}_i$ from $\mathcal{D}_k$, and update the adapter parameters $\theta_{ada\_k}$ from their initialization by minimizing the alignment loss (Section 3.2) via gradient descent, as follows:

$$\theta'_{ada\_k} = \theta_{ada\_k} - \alpha \nabla_{\theta_{ada\_k}} \mathcal{L}_k, \quad \mathcal{L}_k = \|f_{\theta_k}(\mathbf{X}_i) - A_{\theta_{ada\_k}}(f_{\theta_{uni}}(\mathbf{X}_i))\|_2, \tag{1}$$

where $\alpha$ is the step size. For notational simplicity, we consider a single gradient update in the this paper, while using multiple gradient updates is a straightforward extension (See Appendix B). Clearly, we can assume that the effectiveness of adapter training is measured by the test performance of $A_{\theta'_{ada\_k}}(f_{\theta_{uni}}(\cdot))$ with respect to task $k$. Thus, we further simulate the test process of trained adapters and set the meta-objective to improving the performance across all tasks. Concretely, We sample a batch inputs $\mathbf{X}'_i$ from $\mathcal{D}'_k$ and perform inference using $A_{\theta'_{ada\_k}}(f_{\theta_{uni}}(\cdot))$ to get the corresponding features. Similarly, we adopt the feature alignment loss as metric of test performance, and then perform one step meta-update on $\{\lambda_k\}_{k=1}^K$ via gradient descent as follows:

$$\lambda \leftarrow \lambda - \beta \nabla_\lambda \sum_{k=1}^K \mathcal{L}'_k, \quad \mathcal{L}'_k = \|f_{\theta_k}(\mathbf{X}'_i) - A_{\theta'_{ada\_k}}(f_{\theta_{uni}}(\mathbf{X}'_i))\|_2, \tag{2}$$

where $\beta$ is the meta step size. At this point, a meta-update cycle is completed, and the iteration is repeated until convergence. Thus, we leverage lightweight episodes of adapter training and testing to progressively improve future adapter training. Ultimately, we obtain an optimal $\{\lambda_k\}_{k=1}^K$ and corresponding $\theta_{uni}$, such that small updates of adapter based on this can produce large improvements on task performance. Finally, we formulate the bi-level meta-objective as follows:

$$\min_{\{\lambda_1, \lambda_2, ..., \lambda_K\}} \sum_{k=1}^K \sum_{\mathbf{X}'_i \in \mathcal{D}'_k} \|f_{\theta_k}(\mathbf{X}'_i) - A_{\theta^*_{ada\_k}(\theta_{uni})}(f_{\theta_{uni}}(\mathbf{X}'_i))\|_2$$

$$\text{s.t. } \theta^*_{ada\_k}(\theta_{uni}) = \arg\min_{\theta_{ada\_k}} \sum_{\mathbf{X}_i \in \mathcal{D}_k} \|f_{\theta_k}(\mathbf{X}_i) - A_{\theta_{ada\_k}}(f_{\theta_{uni}}(\mathbf{X}_i))\|_2, \, k = 1, 2, ..., K, \tag{3}$$

where $\theta^*_{ada\_k}(\theta_{uni})$ represents the optimized adapter parameters given the united model $\theta_{uni}$. Notably, $\theta_{uni}$ appears inside both $\mathcal{L}_k$ and $\mathcal{L}'_k$, which requires computing second-order derivatives for the meta-update of $\{\lambda_k\}_{k=1}^K$ and thus introduces a significant computational burden. To address this, a technique from FOMAML (Nichol et al., 2018) can be applied, which approximates the update by ignoring the second-order terms and using only first-order derivatives, thereby alleviating the computational cost. In this paper, this approximation is employed only for relatively large models such as GPT-2 and ViT-L/14. In addition, our method is easy to train, as $\{\lambda_k\}_{k=1}^K$ only consists of $K$ values and is easy to converge. More detailed runtime comparisons are provided in the Appendix.

## 4 EXPERIMENT

In this section, we conduct a series of experiments to thoroughly validate our method. Due to page limitations, we defer the hyperparameter analysis, details of experimental setup, and additional results to the Appendix.

### 4.1 MERGING VISION MODELS.

**Datasets and Models.** We follow the most widely adopted model merging setting (Yang et al., 2024d; Huang et al., 2024; Chen and Kwok, 2024; Shen et al., 2024). Specifically, we employ ViT-B/32 and ViT-L/14, two variants of CLIP (Radford et al., 2021) visual encoders, as the pre-trained models. We merge eight fine-tuned models, each fully finetuned on a downstream image classification task: SUN397 (Xiao et al., 2016), Cars (Krause et al., 2013), RESISC45 (Cheng

Table 1: Multi-task performance of merged ViT-B/32 models across eight image classification tasks.

| Methods | SUN397 | Cars | RESISC45 | EuroSAT | SVHN | GTSRB | MNIST | DTD | Avg Acc |
|---|---|---|---|---|---|---|---|---|---|
| Pretrained Model | 62.3 | 59.7 | 60.7 | 45.5 | 31.4 | 32.6 | 48.5 | 43.8 | 48.0 |
| Finetuned Model | 75.3 | 77.7 | 96.1 | 99.7 | 97.5 | 98.7 | 99.7 | 79.4 | 90.5 |
| Multi-task Learning | 73.9 | 74.4 | 93.9 | 98.2 | 95.8 | 98.9 | 99.5 | 77.9 | 88.9 |
| Weight Averaging (Wortsman et al., 2022a) | 65.3 | 63.4 | 71.4 | 71.7 | 64.2 | 52.8 | 87.5 | 50.1 | 65.8 |
| Fisher Merging (Matena and Raffel, 2022) | 68.6 | 69.2 | 70.7 | 66.4 | 72.9 | 51.1 | 87.9 | 59.9 | 68.3 |
| RegMean (Jin et al., 2023) | 65.3 | 63.5 | 75.6 | 78.6 | 78.1 | 67.4 | 93.7 | 52.0 | 71.8 |
| Task Arithmetic (Ilharco et al., 2023) | 63.8 | 62.1 | 72.0 | 77.6 | 74.4 | 65.1 | 94.0 | 52.2 | 70.1 |
| Ties-Merging (Yadav et al., 2023) | 64.8 | 62.9 | 74.3 | 78.9 | 83.1 | 71.4 | 97.6 | 56.2 | 73.6 |
| AdaMerging (Yang et al., 2024d) | 64.5 | 68.1 | 79.2 | 93.8 | 87.0 | 91.9 | 97.5 | 59.1 | 80.1 |
| AdaMerging++ (Yang et al., 2024d) | 66.6 | 68.3 | 82.2 | 94.2 | 89.6 | 89.0 | 98.3 | 60.6 | 81.1 |
| Surgery (Yang et al., 2024b) | 69.8 | 71.0 | 88.9 | 98.1 | 91.7 | 96.5 | 98.8 | 73.6 | 86.1 |
| Pareto Merging (Chen and Kwok, 2025) | 72.1 | 73.7 | 88.8 | 97.5 | 92.2 | 97.5 | 99.0 | 66.1 | 85.9 |
| **MetaMerging** (Ours) | 74.2 | 71.9 | 92.0 | 99.4 | 97.1 | 98.1 | 99.6 | 64.9 | 87.2 |

Table 2: Multi-task performance of merged ViT-L/14 models across eight image classification tasks.

| Methods | SUN397 | Cars | RESISC45 | EuroSAT | SVHN | GTSRB | MNIST | DTD | Avg Acc |
|---|---|---|---|---|---|---|---|---|---|
| Pretrained Model | 66.8 | 77.7 | 71.0 | 59.9 | 58.4 | 50.5 | 76.3 | 55.3 | 64.5 |
| Finetuned Model | 82.3 | 92.4 | 97.4 | 100 | 98.1 | 99.2 | 99.7 | 84.1 | 94.2 |
| Multi-task Learning | 80.8 | 90.6 | 96.3 | 96.3 | 97.6 | 99.1 | 99.6 | 84.4 | 93.5 |
| Weight Averaging (Wortsman et al., 2022a) | 72.1 | 81.6 | 82.6 | 91.9 | 78.2 | 70.7 | 97.1 | 62.8 | 79.6 |
| Fisher Merging (Matena and Raffel, 2022) | 69.2 | 88.6 | 87.5 | 93.5 | 80.6 | 74.8 | 93.3 | 70.0 | 82.2 |
| RegMean (Jin et al., 2023) | 73.3 | 81.8 | 86.1 | 97.0 | 88.0 | 84.2 | 98.5 | 60.8 | 83.7 |
| Task Arithmetic (Ilharco et al., 2023) | 74.1 | 82.1 | 86.7 | 93.8 | 87.9 | 86.8 | 98.9 | 65.6 | 84.5 |
| Ties-Merging (Yadav et al., 2023) | 76.5 | 85.0 | 89.3 | 95.7 | 90.3 | 83.3 | 99.0 | 68.8 | 86.0 |
| AdaMerging (Yang et al., 2024d) | 79.0 | 90.3 | 90.8 | 96.2 | 93.4 | 98.0 | 99.0 | 79.9 | 90.8 |
| AdaMerging++ (Yang et al., 2024d) | 79.4 | 90.3 | 91.6 | 97.4 | 93.4 | 97.5 | 99.0 | 79.2 | 91.0 |
| Surgery (Yang et al., 2024b) | 80.3 | 90.8 | 94.3 | 98.2 | 94.1 | 98.7 | 99.2 | 82.5 | 92.3 |
| Pareto Merging (Chen and Kwok, 2025) | 80.6 | 91.7 | 92.0 | 98.5 | 96.1 | 99.0 | 99.1 | 80.6 | 92.2 |
| **MetaMerging** (Ours) | 82.1 | 90.6 | 97.2 | 99.7 | 97.9 | 98.9 | 99.7 | 80.9 | 93.4 |

et al., 2017), EuroSAT (Helber et al., 2019), SVHN (Yuval, 2011), GTSRB (Stallkamp et al., 2011), MNIST (LeCun, 1998), and DTD (Cimpoi et al., 2014). We report the final performance using classification accuracy. Dataset statistics and preprocessing details are provided in the Appendix.

**Baselines.** We compare our method with multiple baseline model merging method, ranging from classic merging methods to recent advanced methods, including: Weight Averaging (Wortsman et al., 2022a), Fisher Merging (Matena and Raffel, 2022), RegMean (Jin et al., 2023), Task Arithmetic (Ilharco et al., 2023), Ties-Merging (Yadav et al., 2023), AdaMerging (Yang et al., 2024d), AdaMerging++ (Yang et al., 2024d), Surgery (Yang et al., 2024b), and Pareto Merging (Chen and Kwok, 2025). For reference, we also evaluate the pretrained model and the individual fine-tuned models, serving as the upper and lower performance bounds, respectively. In addition, we include multi-task learning as a baseline, in which the multi-task model is obtained by jointly training on all downstream datasets (Zhang and Yang, 2021; Vandenhende et al., 2021).

**Main results.** Table 1 and Table 2 present the multi-task performance of different merging methods on ViT-B/32 and ViT-L/14, respectively. We observe that MetaMerging achieves consistently strong performance across all tasks and model scales. Compared to traditional merging methods without additional training, MetaMerging delivers substantial improvements, often by a margin of 10-20% in average accuracy. Furthermore, even when compared with recent advanced methods that incorporate trainable modules to enhance merging, MetaMerging still surpasses them, demonstrating the effectiveness of our meta-learning based coefficient optimization and adapter training. Moreover, our method achieves performance competitive with multi-task learning without requiring extensive training and access to labeled data.

## 4.2 MERGING LANGUAGE MODELS.

**Datasets and Models.** Following the FusionBench benchmark (Tang et al., 2024a), we evaluate model merging in the NLP domain by merging GPT-2 (Radford et al., 2019) models fine-tuned on seven text classification tasks from GLUE (Wang et al., 2018), which is a popular multi-task benchmark of NLP. Performance on all tasks is measured by accuracy. The dataset statistics and preprocessing details are provided in the Appendix.

Table 3: Multi-task performance of merged GPT-2 models across seven text classification tasks.

| Method | CoLA | SST-2 | MRPC | QQP | MNLI | QNLI | RTE | Avg. |
|---|---|---|---|---|---|---|---|---|
| Pretrained model | 30.8 | 50.9 | 31.4 | 63.2 | 33.3 | 49.2 | 52.7 | 44.5 |
| Finetuned model | 76.8 | 91.2 | 80.4 | 89.6 | 82.1 | 88.3 | 65.3 | 82.0 |
| Weight Averaging (Wortsman et al., 2022a) | 55.0 | 52.5 | 51.0 | 76.7 | 55.1 | 57.6 | 44.8 | 56.1 |
| Fisher Merging (Matena and Raffel, 2022) | 54.8 | 64.7 | 39.5 | 81.5 | 58.0 | 63.3 | 49.1 | 58.7 |
| AdaMerging (Yang et al., 2024d) | 49.7 | 86.7 | 37.5 | 71.2 | 54.8 | 65.3 | 52.1 | 59.6 |
| RegMean (Jin et al., 2023) | 61.7 | 79.7 | 65.4 | 78.8 | 70.4 | 69.7 | 56.0 | 68.8 |
| Task Arithmetic (Ilharco et al., 2023) | 68.7 | 83.6 | 69.6 | 81.8 | 68.6 | 70.5 | 47.3 | 70.0 |
| Ties-Merging (Yadav et al., 2023) | 68.4 | 81.8 | 68.4 | 82.4 | 71.4 | 69.6 | 47.7 | 70.0 |
| **MetaMerging** (Ours) | 71.8 | 90.8 | 79.4 | 78.8 | 69.0 | 82.5 | 66.2 | 76.9 |

**Baselines.** We compare our method with baseline model merging method including: Weight Averaging (Wortsman et al., 2022a), Fisher Merging (Matena and Raffel, 2022), RegMean (Jin et al., 2023), Task Arithmetic (Ilharco et al., 2023), Ties-Merging (Yadav et al., 2023), and AdaMerging (Yang et al., 2024d). Consistent with the experiments on vision models, we also include the pretrained model and fine-tuned model as references.

**Main results.** Table 3 present the results of merging our merging experiment. It can be seen that MetaMerging shows significant performance improvement compare to baseline methods. This demonstrates that our method is applicable not only to vision models but also to language models, and can be extended to various neural network architectures, further verifying the generality and scalability of our method.

## 4.3 MECHANISM ANALYSIS OF METAMERGING.

In this subsection, we aim to investigate the underlying mechanism of MetaMerging and provide insights into how it works. Recall the process of MetaMerging, the core technique is the meta-learning-based optimization of merging coefficient. The merging coefficient $\{\lambda_k\}_{k=1}^K$ control the balance across tasks and shape unified model, which governs the subsequent adapter training and task performance. Firstly, we conduct experiments by applying the same adapter training procedure to unified models obtained from baseline methods and evaluate their performance, as shown in Figure 4. From the results, we observe that

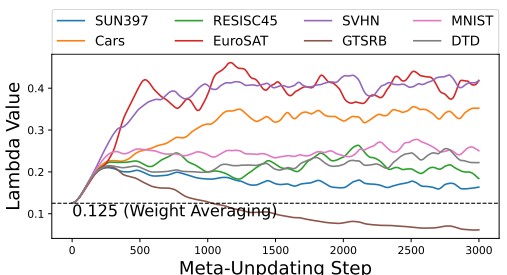

Figure 5: Model merging coefficients $\{\lambda_k\}_{k=1}^K$ change with respect to training steps on merging ViT-B/32 models. Each curve represents the change process of the coefficient $\lambda_k$ of a task vector $T_k$ ($k \in \{1, 2, \ldots, K\}$).

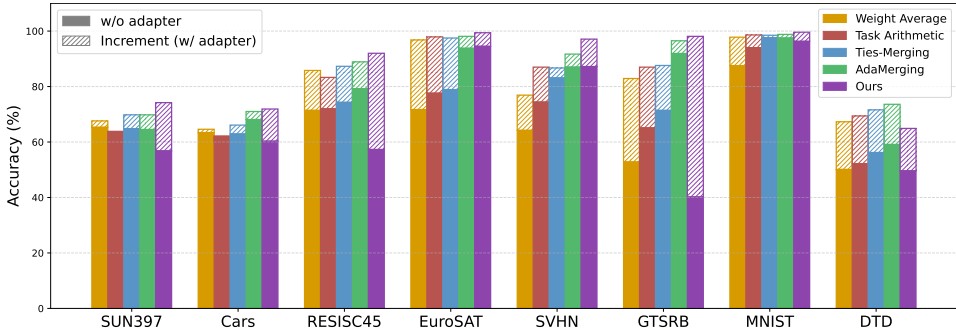

Figure 4: Incremental performance gains from task-specific adapters across unified models merged by different methods (merging ViT-B/32 models for 8 vision tasks). Figure 1 in Section 1 presents the average accuracy version of this chart. More results in Table 5 and Table 6 in Appendix

the effectiveness of the additional adapters depends not only on the underlying unified model but also varies significantly across different tasks. This suggests that the characteristics of a task may determine that its performance tend to depends on the unified model or adapters . In other words, for some tasks, most of the knowledge may come from the shared representation, while other tasks may rely more heavily on task-specific knowledge. For instance, in tasks such as GTSRB and RESISC45, the addition of adapters leads to substantial performance improvements, suggesting a significant knowledge gap in the unified model that necessitates task-specific adapters. In contrast, for tasks such as Cars, SVHN, and MNIST, the performance gains from adapters are much smaller, suggesting that the knowledge from task vectors in the unified model is largely sufficient. An intuitive idea is to increase the weight of tasks that do not require adapters and reduce the weight of others, allowing the adapters to focus on compensating for tasks that need additional task-specific knowledge. To investigate whether MetaMerging optimize the coefficients in this manner, we trace the change process of $\{\lambda_k\}_{k=1}^{K}$ during the meta-learning procedure, as presented in Figure 5. It can be seen that the coefficients $\lambda$ for Cars, SVHN, and MNIST progressively grow and those for GTSRB and RESISC45 diminish during the procedure, consistent with our hypothesis above. This provides an interpretable insight, showing that our method can automatically adapt to the unique characteristics of each task, thereby achieving a balance and optimal performance across all tasks.

On the other hand, we investigate whether the unified model learned by meta-learning has sufficient generalization capacity to be effectively adapted to downstream tasks. Figure 6 shows a comparison of the loss curves when training task-specific adapters on the unified model obtained by our method versus Weight Averaging. It can be observed that task-specific adapters trained with our method converge

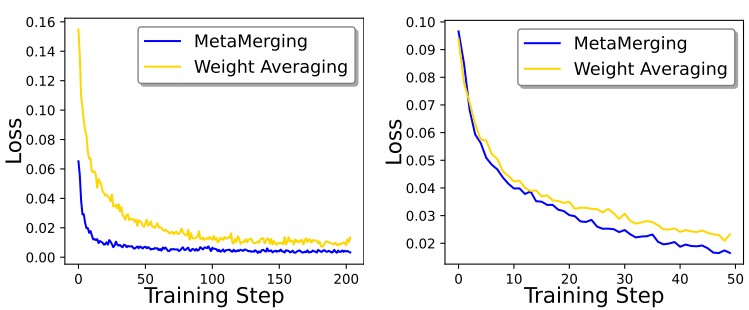

Figure 6: Loss curve when training adapters for SVHN and RESISC45 tasks (another 6 tasks in Appendix), based on the unified model obtained by our method and Weight Averaging.

more rapidly and reach lower loss values, even if the initial point may does not start from a relatively low loss. This indicates that our unified model provides better initialization and greater generalizability for downstream adaptation. Overall, these results further demonstrate both the effectiveness of our method and its interpretability.

### 4.4 Efficiency Analysis of MetaMerging.

We examine the storage and computational costs of MetaMerging. The total number of retained parameters consists only of the single unified model and $K$ task-specific adapters. Since the scale of the adapters is negligible compared to a single encoder model, our method is storage efficient. Moreover, our meta-learning algorithm finds a good initialization for the unified model, enabling adapter training fast and easy to converge, which is discussed above and verified by our experiments. Empirically, our adapter training requires only 1 epoch on the validation dataset, whereas multi-task learning typically requires 10 or more epochs. Consequently, we conduct experiment to compare MetaMerging with baseline methods and multi-task learning, recording the runtime of each merging method. The comparison of parameter counts and runtime is shown in Table 4.

Table 4: The storage and computation efficiency of MetaMerging.

| The total number of parameters | |
| --- | --- |
| ViT-B/32 model | 92,185,089 |
| ViT-L/14 model | 303,179,776 |
| our adapter | 99,136 |

| running time on single 4090 GPU | |
| --- | --- |
| Multi-Task Learning | 15h 53m |
| Adamerging | 2h 5m |
| Surgery | 0h 46m |
| Pareto Merging | 2h 37m |
| MetaMerging | 1h 23m |

## 5 CONCLUSION

In this paper, we address the challenges of recent progressive model merging frameworks. We introduce a multi-task model merging framework, MetaMerging, along with a novel meta-learning algorithm to adaptively optimize the merging coefficients of task vectors. Experimental results demonstrate that our method is effective and intuitively interpretable. Additionally, Our method is widely applicable and can be integrated with techniques that focus on processing task vectors, such as sparsifying them or promoting their orthogonality, potentially providing some inspiration for future research on model merging.

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

## A  ADDITIONAL RESULTS

Here, we provide the additional results as a supplement to the experiments in the main text. Table 5 and Table 6 present the detailed results of the incremental performance analysis in Section 4.3, where the same adapter training process is applied to unified models obtained by different merging methods. Figure 7 shows loss curves of adapter training on the other 6 image classification tasks, complementing the results in Figure 6.

Table 5: Multi-task performance when merging ViT-B/32 models on eight tasks.

| Methods | SUN397 | Cars | RESISC45 | EuroSAT | SVHN | GTSRB | MNIST | DTD | Avg Acc |
|---|---|---|---|---|---|---|---|---|---|
| Pretrained model | 62.3 | 59.7 | 60.7 | 45.5 | 31.4 | 32.6 | 48.5 | 43.8 | 48.0 |
| Finetuned model | 75.3 | 77.7 | 96.1 | 99.7 | 97.5 | 98.7 | 99.7 | 79.4 | 90.5 |
| Weight Averaging w/o adapters | 65.3 | 63.4 | 71.4 | 71.7 | 64.2 | 52.8 | 87.5 | 50.1 | 65.8 |
| Task Arithmetic w/o adapters | 63.8 | 62.1 | 72.0 | 77.6 | 74.4 | 65.1 | 94.0 | 52.2 | 70.1 |
| Ties-Merging w/o adapters | 64.8 | 62.9 | 74.3 | 78.9 | 83.1 | 71.4 | 97.6 | 56.2 | 73.6 |
| AdaMerging w/o adapters | 64.5 | 68.1 | 79.2 | 93.8 | 87.0 | 91.9 | 97.5 | 59.1 | 80.1 |
| **MetaMerging** w/o adapters | 56.8 | 60.3 | 57.3 | 94.5 | 87.2 | 40.2 | 96.3 | 49.6 | 67.8 |
| Weight Averaging w/ adapters | 67.6 | 64.6 | 85.8 | 96.8 | 76.9 | 82.9 | 97.8 | 67.3 | 80.0 |
| Task Arithmetic w/ adapters | 63.8 | 59.9 | 83.3 | 97.9 | 87.0 | 87.0 | 98.6 | 69.4 | 80.9 |
| Ties-Merging w/ adapters | 69.8 | 66.1 | 87.3 | 97.5 | 86.7 | 87.6 | 98.5 | 71.6 | 83.1 |
| AdaMerging w/ adapters | 69.8 | 71.0 | 88.9 | 98.1 | 91.7 | 96.5 | 98.8 | 73.6 | 86.1 |
| **MetaMerging** | 74.2 | 71.9 | 92.0 | 99.4 | 97.1 | 98.1 | 99.6 | 64.9 | 87.2 |

Table 6: Multi-task performance when merging ViT-L/14 models on eight tasks.

| Methods | SUN397 | Cars | RESISC45 | EuroSAT | SVHN | GTSRB | MNIST | DTD | Avg Acc |
|---|---|---|---|---|---|---|---|---|---|
| Pretrained model | 66.8 | 77.7 | 71.0 | 59.9 | 58.4 | 50.5 | 76.3 | 55.3 | 64.5 |
| Finetuned model | 82.3 | 92.4 | 97.4 | 100 | 98.1 | 99.2 | 99.7 | 84.1 | 94.2 |
| Weight Averaging w/o adapters | 72.1 | 81.6 | 82.6 | 91.9 | 78.2 | 70.7 | 97.1 | 62.8 | 79.6 |
| Task Arithmetic w/o adapters | 74.1 | 82.1 | 86.7 | 93.8 | 87.9 | 86.8 | 98.9 | 65.6 | 84.5 |
| Ties-Merging w/o adapters | 76.5 | 85.0 | 89.3 | 95.7 | 90.3 | 83.3 | 99.0 | 68.8 | 86.0 |
| AdaMerging w/o adapters | 79.0 | 90.3 | 90.8 | 96.2 | 93.4 | 98.0 | 99.0 | 79.9 | 90.8 |
| **MetaMerging** w/o adapters | 74.7 | 80.1 | 86.9 | 92.8 | 88.6 | 87.2 | 98.9 | 65.9 | 84.4 |
| Weight Averaging w/ adapters | 73.7 | 83.9 | 92.0 | 98.4 | 82.4 | 86.3 | 98.7 | 71.9 | 85.9 |
| Task Arithmetic w/ adapters | 75.7 | 84.4 | 93.1 | 98.8 | 91.3 | 93.4 | 99.1 | 76.1 | 89.0 |
| Ties-Merging w/ adapters | 76.5 | 85.9 | 93.7 | 99.2 | 89.7 | 92.0 | 99.1 | 78.1 | 89.3 |
| AdaMerging w/ adapters | 80.3 | 90.8 | 94.3 | 98.2 | 94.1 | 98.7 | 99.2 | 82.5 | 92.3 |
| **MetaMerging** | 82.1 | 90.6 | 97.2 | 99.7 | 97.9 | 98.9 | 99.7 | 80.9 | 93.4 |

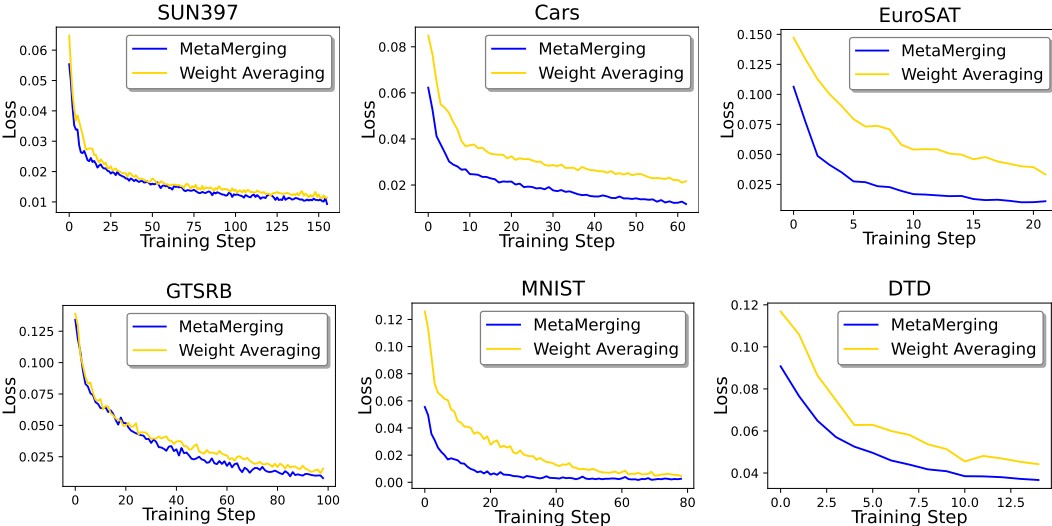

Figure 7: The detailed results of experiments in Figure 6 Section 4.3. We present the additional loss curve of adapter training on 6 tasks.

## B   HYPER-PARAMETER ANALYSIS

We further examine the impact of hyper-parameters (step size $\alpha$ and meta step size $\beta$) in our meta-learning algorithm. As shown in Table 7 and Table 8, we present the performance variety of our method with $\alpha$ and $\beta$ ranging from 1 to 0.001, respectively. It can be observed that too large or too small values of both $\alpha$ and $\beta$ lead to suboptimal performance. By the way, we record the runtime and compare it to multi-task learning.

Additionally, in the meta-learning iteration of our method, the number of inner-step updates for the adapter is a hyperparameter that plays a critical role in the efficiency of the meta-learning process. We evaluate its impact when merging GPT-2 models and report the runtime and average performance in Figure 8 and Table 9.

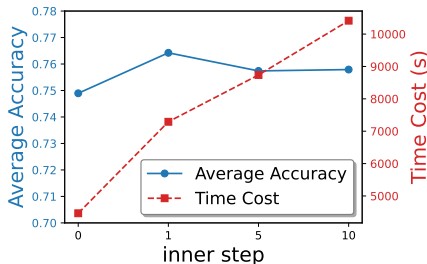

Figure 8: Impact of inner steps when merging GPT-2 models

Table 7: Impact of different step sizes $\alpha$ in meta-learning. Multi-task performance of merging ViT-B/32 models on eight image classification tasks. Training time is measured on a single NVIDIA RTX 4090 GPU.

| Methods | SUN397 | Cars | RESISC45 | EuroSAT | SVHN | GTSRB | MNIST | DTD | Avg Acc | training times |
|---|---|---|---|---|---|---|---|---|---|---|
| Pretrained Model | 62.3 | 59.7 | 60.7 | 45.5 | 31.4 | 32.6 | 48.5 | 43.8 | 48.0 | None |
| Finetuned Model | 75.3 | 77.7 | 96.1 | 99.7 | 97.5 | 98.7 | 99.7 | 79.4 | 90.5 | None |
| Multi-task Learning | 73.9 | 74.4 | 93.9 | 98.2 | 95.8 | 98.9 | 99.5 | 77.9 | 88.9 | 15h 53m 16s |
| step size 1 | 73.53 | 70.00 | 92.57 | 99.52 | 97.31 | 98.53 | 99.50 | 62.98 | 86.74 | 1h 36m 52s |
| step size 0.1 | 74.34 | 70.46 | 92.44 | 99.52 | 97.36 | 97.72 | 99.57 | 64.84 | 87.03 | 1h 42m 58s |
| step size 0.01 | 74.55 | 70.40 | 92.51 | 99.52 | 97.31 | 97.14 | 99.54 | 63.83 | 86.85 | 1h 35m 27s |
| step size 0.001 | 74.45 | 70.25 | 92.56 | 99.52 | 97.34 | 97.40 | 99.53 | 63.78 | 86.85 | 1h 37m 16s |

Table 8: Impact of different meta step sizes $\beta$ in meta-learning. Multi-task performance of merging ViT-B/32 models on eight image classification tasks. Training time is measured on a single NVIDIA RTX 4090 GPU.

| Methods | SUN397 | Cars | RESISC45 | EuroSAT | SVHN | GTSRB | MNIST | DTD | Avg Acc | training times |
|---|---|---|---|---|---|---|---|---|---|---|
| Pretrained Model | 62.3 | 59.7 | 60.7 | 45.5 | 31.4 | 32.6 | 48.5 | 43.8 | 48.0 | None |
| Finetuned Model | 75.3 | 77.7 | 96.1 | 99.7 | 97.5 | 98.7 | 99.7 | 79.4 | 90.5 | None |
| Multi-task Learning | 73.9 | 74.4 | 93.9 | 98.2 | 95.8 | 98.9 | 99.5 | 77.9 | 88.9 | 15h 53m 16s |
| meta step size 1 | 72.72 | 65.75 | 94.98 | 99.59 | 97.36 | 96.66 | 99.36 | 59.52 | 85.74 | 1h 29m 42s |
| meta step size 0.1 | 74.39 | 70.66 | 91.44 | 99.56 | 97.37 | 97.69 | 99.52 | 66.38 | 87.13 | 1h 31m 34s |
| meta step size 0.01 | 74.34 | 70.46 | 92.44 | 99.52 | 97.36 | 97.72 | 99.57 | 64.84 | 87.03 | 1h 23m 34s |
| meta step size 0.001 | 74.46 | 69.03 | 93.11 | 99.30 | 97.26 | 98.60 | 99.50 | 62.18 | 86.68 | 1h 57m 4s |

Table 9: Impact of different inner steps in meta-learning. Multi-task performance when merging GPT-2 models on seven text classification tasks. The training time is measured on distributed training with 4 NVIDIA RTX 4090 GPUs.

| Method | CoLA | SST-2 | MRPC | QQP | MNLI | QNLI | RTE | Avg ACC | training time (s) |
|---|---|---|---|---|---|---|---|---|---|
| Pretrained model | 30.8 | 50.9 | 31.4 | 63.2 | 33.3 | 49.2 | 52.7 | 44.5 | None |
| Finetuned model | 76.8 | 91.2 | 80.4 | 89.6 | 82.1 | 88.3 | 65.3 | 82.0 | None |
| inner steps 0 | 69.3 | 89.6 | 76.2 | 78.2 | 60.1 | 84.7 | 66.2 | 74.9 | 6465.49 |
| inner steps 1 | 71.5 | 88.4 | 78.9 | 78.8 | 67.5 | 84.0 | 65.8 | 76.4 | 7291.45 |
| inner steps 5 | 69.5 | 90.8 | 78.4 | 82.9 | 61.9 | 80.1 | 66.5 | 75.7 | 8736.94 |
| inner steps 10 | 71.4 | 88.3 | 79.4 | 82.8 | 63.0 | 81.3 | 64.4 | 75.8 | 10413.92 |

## C   EXPERIMENT SETUP DETAILS

**Data Preprocessing and Splitinng.** Following the (Ilharco et al., 2023) and (Huang et al., 2024) for vision and language model merging experiments, we split each dataset into training, validation,

and test sets. In our meta-learning algorithm (Algorithm 1), we use the training set as $\mathcal{D}_k$, the validation set as the mate-test dataset $\mathcal{D}'_k$, and the test set as the final evaluation of the performance of MetaMerging. The data preprocessing strategy also follows the previous work (Ilharco et al., 2023).

**Training setting and Computational Environment.** In our experiments, we adopt an adapter architecture as follows

$$A_\theta(\mathbf{h}) = \mathbf{h} - Linear(ReLU(Linear(h)))$$

where $Linear(\mathbf{h}) = \mathbf{W} \cdot \mathbf{h} + \mathbf{b}$ is a simple linear layer, and the hidden dimension inside the adapter, referred to as the rank, determines the scale of the adapter. In all experiments, we set the adapter rank to 64, while the token feature dimension is 768. We set inner step size $\alpha$ to 0.1 and $\beta$ to 0.01 and perform one gradient update in the inner loop. The sensitivity of these hyperparameters is analyzed in Appendix B. For merging ViT-B/32 models, we run the experiments on a single NVIDIA RTX 4090 GPU; for merging ViT-L/14 and GPT-2 models, we run the experiments in parallel on 4 NVIDIA RTX 4090 GPUs. For the runtime experiments, the adapter is trained for one epoch in stage (2) of our method, and for multi-task learning, it is trained for ten epochs.

# D DATASETS DETAILS

Here, we summarize the details of the datasets used in our experiments. All datasets used in our experiments are publicly accessible.

**Image classification datasets** are illustrated in Table 10. All images are resized to 224×224 for ViT input. Standard data augmentations are applied (random crop, horizontal flip, normalization).

Table 10: Image Classification Datasets for ViT Merge Experiments

| Dataset | #Train | #Test | #Classes | Image Size / Notes |
|---|---|---|---|---|
| SUN397 | 108,754 | 8,000 | 397 | Natural scenes, resized to 224×224 |
| Cars (Stanford Cars) | 8,144 | 8,041 | 196 | Car models, resized to 224×224 |
| RESISC45 | 22,500 | 9,000 | 45 | Remote sensing images, resized to 224×224 |
| EuroSAT | 21,600 | 5,400 | 10 | Satellite images, resized to 224×224 |
| SVHN | 73,257 | 26,032 | 10 | Street view house numbers, resized to 32×32 → 224×224 |
| GTSRB | 39,209 | 12,630 | 43 | Traffic signs, resized to 32–250×32–250 → 224×224 |
| MNIST | 60,000 | 10,000 | 10 | Handwritten digits, resized to 28×28 → 224×224 |
| DTD | 3,760 | 1,880 | 47 | Texture dataset, resized to 224×224 |

**Text classification datasets** are illustrated in 11. We tokenize all datasets with the GPT-2 tokenizer and pad tokens to the maximum sequence length.

Table 11: Text Classification Datasets for GPT-2 Merge Experiments

| Dataset | #Train | #Validation | #Test | Task Type |
|---|---|---|---|---|
| CoLA | 8,551 | 1,043 | 1,063 | Linguistic acceptability (binary) |
| SST-2 | 67,349 | 872 | 1,821 | Sentiment analysis (binary) |
| MRPC | 3,668 | 408 | 1,725 | Paraphrase detection (binary) |
| QQP | 363,849 | 40,430 | 391,315 | Question similarity (binary) |
| MNLI | 392,702 | 9,815 / 9,824 | 9,815 / 9,824 | Natural language inference (3-class) |
| QNLI | 104,743 | 5,463 | 5,463 | Question-answer entailment (binary) |
| RTE | 2,490 | 277 | 3,000 | Textual entailment (binary) |
| WNLI | 634 | 71 | 146 | Coreference / entailment (binary) |

# E BASELINE DETAILS

We provide a brief introduction of the baseline method used in our model merging experiments as follows:

**Weight Averaging** (Wortsman et al., 2022a): A simple ensembling approach that averages the parameters of finetuned models. While computationally efficient, it often leads to performance degradation due to conflicting task-specific knowledge.

**Fisher Merging** (Matena and Raffel, 2022): Utilizes the Fisher information matrix to weight model parameters during merging, aiming to preserve important knowledge from each model.

**RegMean** (Jin et al., 2023): Introduces a regularized averaging strategy to mitigate conflicts between models and achieve smoother parameter interpolation.

**Task Arithmetic** (Ilharco et al., 2023): Constructs task vectors from finetuned models and merges them through linear arithmetic with coefficients.

**Ties-Merging** (Yadav et al., 2023): Improves merging stability by resolving parameter conflicts based on tied weights across models.

**AdaMerging** (Yang et al., 2024d): Learns adaptive merging coefficients via lightweight optimization, enabling task-aware parameter integration.

**AdaMerging++** (Yang et al., 2024d): An enhanced version of AdaMerging with refined strategies for coefficient learning and conflict mitigation.

**Surgery** (Yang et al., 2024b): Add task-specific surgery modules (adapters) on unified model to mitigate the "representation bias" of model merging, thereby narrowing the performance gap between model merging and individual finetuned models.

**Pareto Merging** (Chen and Kwok, 2025): Treat model merging as a multi-task optimization problem. By introducing parameter-efficient structures, it generates a Pareto set of merged models, from which users can select according to their preferences or tailor to a specific task.

