# OpenReview forum: "Learn to Merge: Meta-Learning for Adaptive Multi-Task Model Merging"
_ICLR.cc/2026/Conference — Submitted to ICLR 2026_

### Official Review · Reviewer_9SxD · 2025-10-22

**Soundness:** 3
**Presentation:** 2
**Contribution:** 3
**Rating:** 6
**Confidence:** 2

**Summary:**

This paper introduces MetaMerging, a meta-learning-based framework for adaptive model merging in multi-task learning. Traditional model merging methods (e.g., Task Arithmetic, Ties-Merging, Surgery, AdaMerging) often rely on fixed or manually tuned merging coefficients when combining task vectors from fine-tuned models. The proposed method innovatively uses meta-learning to automatically optimize these merging coefficients, improving the generalization of the unified model and facilitating the training of task-specific adapters. Experiments on vision (ViT-B/32, ViT-L/14) and language (GPT-2) models show that MetaMerging achieves higher average accuracy across multiple tasks than prior merging methods.

**Strengths:**

The key contribution—using meta-learning to optimize merging coefficients—is conceptually elegant and fills a gap in existing merging methods.

This paper is well-written.

Figure 3 is easy to follow.

**Weaknesses:**

Although comparisons are made to AdaMerging, Surgery, and Pareto Merging, newer or hybrid model merging approaches (e.g., MoE-based fusion or gradient-space merging methods) are not included. The omission limits the claim of state-of-the-art performance.

Meta learning requires backward transfer that needs the calculation of gradient, even second-order gradient, which is infeasible for large-scale language models, such as Qwen3 32B.

The collection of meta-train and meta-test sets is a challenge for powerful models, such as LLMs.

The improvements over Surgery are limited, while the training and memory requirements seem much higher than Surgery.

Table 4 should include the training time of existing model merging methods.

**Questions:**

See weakness.

---

> ### Author Response · Authors · 2025-11-26
> **Response to Weakness 1 & 2**
>
> Thank you for the valuable and constructive feedback. Below we present our response.
>
>
> **W1**
>
> Thanks for this helpful suggestion.
>
> We have now included comparisons with three recent model-merging approaches: one MoE-based method WEMoE [1], and two gradient-space merging methods DOGE [2] and SuperMerge [3] (when merging 8 ViT-B/32 models). The results are shown below:
>
>
> | Method            | SUN397 | Cars | RESISC45 | EuroSAT | SVHN | GTSRB | MNIST | DTD  | Avg.  |
> |-------------------|--------|------|----------|---------|------|-------|-------|------|------|
> | **WEMOE[1]**          | 74.1   | 77.4 | 93.7     | 99.1    | 96.2 | 98.9  | 99.6  | 76.4 | 89.4 |
> | **MetaMerging (ours)**| 74.2   | 71.9 | 92.0     | 99.4    | 97.1 | 98.1  | 99.6  | 64.9 | 87.2 |
> | **DOGE[2]**          | 70.5   | 74.8 | 88.7     | 94.1    | 91.6 | 95.7  | 98.8  | 72.5 | 85.9 |
> | **SuperMerge[3]**    | 67.9   | 73.4 | 90.7     | 97.6    | 95.1 | 96.2  | 97.9  | 66.4 | 85.7 |
>
>
>
> The results show that our method outperforms DOGE and SuperMerge, though it does not surpass WEMoE.
>
> However, it is important to note that the parameter count of our merged model for 8 tasks is 89.05M, which is close to a single-task ViT-B/32 model. In contrast, the WEMoE model for 8 tasks has 573.96M parameters (refer to Table 8 in [1]), which is over $6\times$ larger, leading to substantially higher memory consumption. This difference arises because our adapter modules are lightweight, whereas WEMoE relies on a significantly heavier MoE architecture.
>
> Recent advanced merging methods often rely on complex architectures or additional training stages. We believe that there is no absolute "SOTA" in model merging, as methods with higher accuracy frequently come with much higher training and parameter costs. Our work instead aims to provide a balanced trade-off between performance and efficiency, which we view as essential for scalable multi-task deployment.
>
>
> [1]ICML'24 Merging Multi-Task Models via Weight-Ensembling Mixture of Experts. https://openreview.net/forum?id=nLRKnO74RB
> [2]ICML'25 Modeling Multi-Task Model Merging as Adaptive Projective Gradient Descent. https://openreview.net/forum?id=EqoKRSR5Pa
> [3]ICML'25 SUPERMERGE: An Approach For Gradient-Based Model Merging. https://openreview.net/forum?id=lIdc5DUplq
>
>
> **W2**
>
> We appreciate this concern.
> Our approach is explicitly designed to make meta-learning feasible for large models through the following aspects:
> (1) In our work, we adopt `first-order meta-learning` for large models, which omits second-order gradients. This reduces computation by 50~70% and avoids the associated memory overhead. This is also the setting used in our main experiments on merging GPT-2 models clarified in Section 3 line 304. We believe that first-order meta-learning is also applicable to Qwen-32B.
> (2) During meta-training, only the lightweight adapter parameters participate in gradient computation. As a result, no backward pass flows through the full LLM backbone, and the memory cost is comparable to that of standard adapter fine-tuning.
> (3) The inner-loop step number (one of the most influential factors determining meta-learning complexity) can be flexibly adjusted to control computational cost. We report in Table 9 that selecting a feasible number of inner steps yields substantial training-time reduction while preserving performance.

---

> ### Author Response · Authors · 2025-11-26
> **Response to Weakness 3 & 4 & 5**
>
> **W3**
>
> We would like to clarify that the downstream task data used for meta-train/meta-test in our work does not require any labels. Our framework only needs raw text to define task-specific inputs, which makes data collection considerably easier for large models.
> This also suggests that synthetic text produced by an LLM could, in principle, be useful for our meta-training tasks, since the method does not rely on annotated supervision.
>
>
>
> **W4**
>
> We measured the peak GPU memory when merging 8 ViT-B/32 models with batch size 128 on a single 4090 GPU. The results are shown below:
>
> | Method               | Peak GPU Memory |
> |---------------------|----------------|
> | Surgery             | 14.2 GB        |
> | MetaMerging (ours)  | 22.3 GB        |
>
>
> From these results, our method uses moderately more memory, but it remains acceptable.
> Additionally, we added the training time comparison in Table 4 of the revised paper (also shown in w5). The time cost of MetaMerging is only slightly higher than Surgery and still far lower than full multi-task training, demonstrating that our method is efficient in practice.
>
> Furthermore, model merging typically considers the performance of each task's fine-tuned model as an upper bound. In this setting, achieving performance close to the upper bound is challenging. MetaMerging improves over Surgery by 1.1% average accuracy, which is a meaningful gain given the difficulty of surpassing the upper-bound performance. Thus, the modest increase in memory and training cost is justified by the performance gain, confirming the feasibility of our approach.
>
>
>
>
>
>
> **W5**
>
> Thanks for the suggestion. We have added the time comparison with baseline methods that need training (see revised paper). And we also present the comparison below (Merging 8 ViT-B/32 models on single 4090 GPU).
>
> | Method            | Time Cost  |
> |------------------|------------|
> | **Adamerging**        | 2h 5m     |
> | **Surgery**           | 0h 46m    |
> | **Pareto Merging**    | 2h 37m    |
> | **MetaMerging (ours)**| 1h 31m    |
> | **Multi-task Learning**        | 15h 53m    |

---

### Official Review · Reviewer_5icd · 2025-10-31

**Soundness:** 3
**Presentation:** 3
**Contribution:** 3
**Rating:** 6
**Confidence:** 2

**Summary:**

This paper investigates the problem of merging multiple fine-tuned models into a unified multi-task model, aiming to enhance efficiency and performance across diverse tasks. The authors propose a meta-learning-based framework, MetaMerging, which adaptively learns optimal merging coefficients for task vectors, moving beyond fixed or heuristic merging strategies. This method allows the unified model to better preserve and leverage task-specific knowledge, facilitating improved adapter training and multi-task generalization. Experimental results on both vision and language benchmarks demonstrate that MetaMerging consistently outperforms conventional model merging techniques in accuracy and training efficiency. The work offers a principled approach for adaptive model consolidation, contributing to the advancement of scalable and robust multi-task learning.

**Strengths:**

**Originality:**

The paper proposes a meta-learning approach for merging multiple fine-tuned models, offering a new way of adaptively learning merging coefficients rather than relying on fixed or manual strategies. This creative combination of meta-learning and multi-task model consolidation addresses a practical gap in existing literature.

**Quality:**

Methodologically, the framework is well-founded and experimentally validated across diverse benchmarks in both vision and language domains. The empirical analysis demonstrates consistent improvements over standard baseline methods, with clear, quantitative results supporting the core claims.

**Clarity:**

The presentation is logical, with well-structured explanations, informative figures, and thorough comparative tables. While dense in the technical sections, the overall narrative is coherent and the methodology is transparent for readers familiar with deep learning.

**Significance:**

By providing a principled solution for scalable model merging, the work has practical significance for resource-efficient deployment of multi-task systems. Its adaptive strategy enables better generalization and performance, contributing meaningfully to the progress of the field and addressing real-world challenges in model consolidation and transfer learning.

**Weaknesses:**

- **Limited Novelty Relative to Existing Adaptive Merging:** Related ideas of adaptive weighting exist in ensemble learning and prior neural model merging. The paper would benefit from clearer differentiation and explicit discussion of how its approach surpasses past adaptive strategies (e.g., with more theoretical justification or unique robustness properties).
- **Generalization Beyond Benchmarks:** The experiments, though thorough on selected datasets and architectures, are limited to standard benchmarks. To strengthen the validity of claims, additional studies on more diverse domains, truly large-scale settings, or real-world multi-task scenarios (with heterogeneous architectures or data) would be valuable.
- **Accessibility and Implementation Details:** The methodology sections (3.1,3.2,3.3) remain technically dense, making it difficult for broader audiences to understand the key concept. I suggest author to provide a brief introduction to basically describe the motivation and purpose and methodology of this section. Being a high-level one for the user to understand the general idea. Then you can describe your detailed procedure or massive computation.

**Questions:**

1. Can the authors provide evidence or commentary on how their meta-merging approach would handle truly large-scale, real-world scenarios with many diverse tasks, different model architectures, or highly imbalanced datasets? What specific challenges or modifications might arise when scaling beyond the standard benchmarks presented?
2. Could the authors clarify how their meta-learning method differs fundamentally from previous adaptive weighting schemes in model merging and ensembling literature? Are there theoretical advantages, unique robustness properties, or empirical tests that distinguish MetaMerging as more than an incremental improvement?

---

> ### Author Response · Authors · 2025-11-26
> **Response to Weakness 1 & 2 & 3**
>
> Thank you for the valuable and constructive feedback. Below we present our response.
>
> **W1**
>
> We appreciate the reviewer's comment regarding differentiation from existing adaptive merging methods. MetaMerging differs fundamentally in its objective and mechanism. Unlike prior adaptive approaches, which typically optimize inference–time weights or post-hoc coefficients, MetaMerging optimizes pre-merge coefficients that directly shape the internal representation space of the unified model. This ensures that the merged model itself, before any downstream adaptation, is better aligned for multiple tasks.
>
> Moreover, our method employs a bilevel meta-learning formulation, where the outer loop updates merging coefficients based on downstream adapter performance, while the inner loop adapts task-specific modules. This bilevel optimization, coupled with automatic handling of conflicting task directions via meta-gradients, is not present in prior adaptive merging or ensemble weighting schemes and provides a principled way to achieve more robust multi-task merging.
>
>
> **W2**
>
> We would like to clarify that the datasets we used are standard benchmarks in the majority of the model merging literature [1][2][3][4][5][6]. These benchmarks are both representative and sufficient to demonstrate the effectiveness and generality of our method. We believe they are diverse enough to reflect real-world multi-task scenarios.
>
> [1]NeurIPS'2024 EMR-Merging: Tuning-Free High-Performance Model Merging. https://openreview.net/forum?id=lYdjzx3DYu
> [2]ICCV'2024 FREE-Merging: Fourier Transform for Efficient Model Merging. https://arxiv.org/pdf/2411.16815
> [3]ICML'2024 Localizing Task Information for Improved Model Merging and Compression. https://openreview.net/forum?id=DWT9uiGjxT
> [4]ICLR'2024 AdaMerging: Adaptive Model Merging for Multi-Task Learning. https://openreview.net/forum?id=nZP6NgD3QY
> [5]ICML'2024 Representation Surgery for Multi-Task Model Merging. https://openreview.net/forum?id=Sbl2keQEML
> [6]fusionbench FusionBench: A Comprehensive Benchmark of Deep Model Fusion. https://openreview.net/forum?id=a0sK0foX3p
>
>
> **W3**
>
> We thank the reviewer for this suggestion. In the revision, we have added a short high-level overview before the formal definitions and algorithmic details (marked blue in the revised paper), clearly explaining describe the motivation and purpose and methodology. We believe this will make the method more accessible to a broader audience without changing the technical content.
>
> Specifically, the inserted summary is as follows:
>
> In this section, we introduce MetaMerging, our framework for merging multiple task-specific models into a single, efficient multi-task model. Starting from a pretrained backbone and its fine-tuned variants. Specifically, we first represent each task by a task vector—the difference between its fine-tuned model and the pretrained model, and combine these vectors into a unified model using a set of merging coefficients. We then attach lightweight adapters for each task on top of this unified model. The key idea is to meta-learn the merging coefficients so that, after a few steps of adapter training on unlabeled data, the unified model plus adapters mimic the original fine-tuned models as well as possible. In Section 3.1, ...

---

> ### Author Response · Authors · 2025-11-26
> **Response to Question 1 & 2**
>
> **Q1**
>
> We appreciate the reviewer's question about how meta-merging might behave in truly large-scale, real-world settings. Our current experiments, in line with main existing work on model merging, focus on the pretrain-finetune paradigm: all task-specific models are finetuned from a shared pretrained checkpoint and therefore share the same architecture.
>
> For models with different architectures, our method in its current form assumes parameter alignment across models, so a direct application is not possible. A natural extension would be to first bring all task models into a common representation space and then perform merging there. One concrete approach is to distill heterogeneous models into a shared backbone and then apply our meta-merging procedure on the distilled models. Alternatively, one could meta-learn both the merging coefficients and adapters jointly. We view this as an important and practically relevant direction for future work.
>
> Regarding a large number of diverse tasks, our benchmark is designed to include tasks that differ in domain and label structure, and we believe it captures many aspects of realistic multi-task scenarios. When involving hundreds or thousands of tasks, we could introduce sparsity or grouping in the merging coefficients so that only related tasks strongly influence each other. For highly imbalanced datasets, our current experiments already include tasks with differing dataset sizes, but not extreme imbalance. In principle, our meta-learning objective can be adapted to such cases by reweighting tasks in the meta-loss, for example, upweighting low-resource tasks or using inverse-frequency weighting. We expect these modifications to improve robustness, and we consider systematic evaluation under such interesting settings for future work.
>
>
>
>
> **Q2**
>
> We thank the reviewer for this question and will clarify in the revision how our approach differs from prior adaptive weighting methods for merging. At a high level, MetaMerging is not a heuristic, one-shot weighting rule, but a bi-level meta-learning framework in which the merging coefficients are optimized end-to-end for post-adaptation performance. Specifically:
>
> - Most adaptive schemes operate in output space or use fixed, heuristic rules in parameter space. MetaMerging instead meta-learns coefficients that directly combine task vectors in parameter space, with these coefficients optimized for downstream performance rather than chosen a priori.
>
> - Our objective is not the immediate performance of a merged model. Instead, we use a bilevel meta-learning objective in which coefficients are optimized based on future adapter improvement: after a few inner-loop adapter updates, the adapted merged model should closely match the original fine-tuned models. This "optimize for post-adaptation performance" view is different from prior adaptive weighting approaches.
>
> - The outer-loop coefficients and inner-loop adapter updates are coupled: gradients for the coefficients pass through the adapter training steps. Thus, the learned merge is explicitly shaped to support fast adaptation on all tasks, which naturally discourages producing strong negative transfer. Thus, this leads to more robust performance than standard adaptive weighting baselines.

---

### Official Review · Reviewer_uyTM · 2025-10-31

**Soundness:** 2
**Presentation:** 3
**Contribution:** 2
**Rating:** 2
**Confidence:** 4

**Summary:**

This paper proposes a model merging technique with meta-learning of merging coefficients for training task-specific adapters. The study followed closely on how model merging problems and experiments have been defined and studied. The writing and organization of the paper are good overall.

**Strengths:**

1) The work investigates the method & impact of task-specific adapters for a multi-task model, and shows that the proposed method could slightly perform better than the compared works.
2) The method is clearly described for meta-learning and adapters.
3) The experiment is followed model merging literature closely.

**Weaknesses:**

1) Figures 2 and 3 can be improved; it is difficult to understand without reading the entire paper.
2) Missing SOTA comparison with the data-less model merging method, WUDI merging.
3) The performance improvement against SOTA is marginal for VIT, despite requiring data, gradient, and additional adapters.
4) In the NLP experiment, the proposed method is only compared with weak baselines (all before the end of 2023), and a relatively small GPT2 was used (recent model merging at least uses llama2/3).

**Questions:**

1) Table 4 should include the running times for all methods compared.
2) It would be good to evaluate out-of-domain generalisation

---

> ### Author Response · Authors · 2025-11-26
> **Response to Weakness 1 & 2 & 3 & 4**
>
> Thank you for the thoughtful feedback. We provide our detailed response as follows:
>
> **W1**
>
> Thanks for pointing this out. To improve clarity, we will try to revise these two figures to make them more clear. Also, we will improve the corresponding textual description for further facilitate understanding.
>
>
> **W2**
>
> Here we present a comparison with WUDI-Merging. The two tables correspond to merging 8 ViT-B/32 models and 8 ViT-L/14 models, respectively.
>
> | Method             | SUN397 | Cars | RESISC45 | EuroSAT | SVHN | GTSRB | MNIST | DTD  | Avg. |
> |--------------------|--------|------|----------|---------|------|-------|-------|------|------|
> | WUDI-Merging [1]   | 71.1   | 71.0 | 85.7     | 95.6    | 94.2 | 94.7  | 99.5  | 69.7 | 85.2 |
> | **MetaMerging (ours)** | 74.2   | 71.9 | 92.0     | 99.4    | 97.1 | 98.1  | 99.6  | 64.9 | 87.2 |
>
>
> | Method             | SUN397 | Cars | RESISC45 | EuroSAT | SVHN | GTSRB | MNIST | DTD  | Avg. |
> |--------------------|--------|------|----------|---------|------|-------|-------|------|------|
> | WUDI-Merging [1]      | 81.0   | 91.0 | 94.2     | 99.2    | 96.3 | 98.1  | 99.6  | 81.2 | 92.6 |
> | **MetaMerging (Ours)** | 82.1   | 90.6 | 97.2     | 99.7    | 97.9 | 98.9  | 99.7  | 80.9 | 93.4 |
>
>
> From the results, our method consistently outperforms WUDI-Merging across most tasks and on average. Although WUDI-Merging is a data-free approach, it still requires an optimization procedure via gradient descent based on a predefined loss function (see Equation 21 in [1]). We will include the baseline and results in the revised version.
>
> [1]ICML'2025 Whoever Started the interference Should End It: Guiding Data-Free Model Merging via Task Vectors. https://openreview.net/forum?id=xR9msNaREW
>
> **W3**
>
> Despite requiring additional components, our approach remains a feasible and effective merging strategy for high performance. The adapter module we introduce is lightweight, and training is efficient, which is substantially faster than multi-task training. Moreover, nearly all recent high-performance merging methods [1][2][3][4] introduce auxiliary modules or require access to data, meaning the use of additional components is now the norm rather than an overhead.
>
> It is also important to note that model merging typically considers the performance of individually fine-tuned models as an upper bound, making further improvements particularly challenging. In this context, MetaMerging improves upon Surgery by +1.1% on average, which represents a meaningful gain given the difficulty of approaching the upper-bound performance. Thus, the modest increase in memory and training cost is well justified by the performance improvement, supporting the overall practicality and scalability of our method.
>
>
> [1]ICML'2025 Pareto Merging: Multi-Objective Optimization for Preference-Aware Model Merging. https://openreview.net/forum?id=D7qRwx6BOS
> [2]NeurIPS'2024 Twin-Merging: Dynamic Integration of Modular Expertise in Model Merging. https://openreview.net/forum?id=81YIt63TTn
> [3]NeurIPS'2024 EMR-Merging: Tuning-Free High-Performance Model Merging. https://openreview.net/forum?id=lYdjzx3DYu
> [4]ICML'2024 Merging Multi-Task Models via Weight-Ensembling Mixture of Experts. https://openreview.net/forum?id=nLRKnO74RB
>
> **W4**
>
> We thank the reviewer for pointing out the baseline selection in the NLP setting. While GPT-2 is smaller than recent LLaMA-based LMs, we note that existing model-merging works in NLP are still predominantly based on GPT-2 and RoBERTa [1][2][3]. Moreover, most recent merging studies [1-5] focus primarily on ViT-B/32 and ViT-L/14 as evaluation platforms, with relatively few investigations on language models. Therefore, GPT-2 remains a suitable benchmark for demonstrating the effectiveness of our method in NLP, which motivated our choice.
>
> We agree that extending evaluation to LLaMA-scale models is valuable for demonstrating generality, and we consider this direction highly promising. While full-scale LLaMA merging experiments are beyond the timeline of this rebuttal, in future work, we plan to extend our experiments to larger models such as LLaMA 2/3.
>
> [1]NeurIPS'2024 EMR-Merging: Tuning-Free High-Performance Model Merging. https://openreview.net/forum?id=lYdjzx3DYu
> [2]NeurIPS'2024 Twin-Merging: Dynamic Integration of Modular Expertise in Model Merging. https://openreview.net/forum?id=81YIt63TTn
> [3]fusionbench FusionBench: A Comprehensive Benchmark of Deep Model Fusion. https://openreview.net/forum?id=a0sK0foX3p
> [4]ICML'2025 Whoever Started the interference Should End It: Guiding Data-Free Model Merging via Task Vectors. https://openreview.net/forum?id=xR9msNaREW
> [5]ICML'2025 Pareto Merging: Multi-Objective Optimization for Preference-Aware Model Merging. https://openreview.net/forum?id=D7qRwx6BOS

---

> ### Author Response · Authors · 2025-11-26
> **Response to Question 1 & 2**
>
> **Q1**
>
> Thanks for the suggestion. We have added the time comparison with baseline methods that need training (see revised paper). And we also present the comparison below (Merging 8 ViT-B/32 models on single 4090 GPU).
>
> | Method            | Time Cost  |
> |------------------|------------|
> | **Adamerging**        | 2h 5m     |
> | **Surgery**           | 0h 46m    |
> | **Pareto Merging**    | 2h 37m    |
> | **MetaMerging (ours)**| 1h 31m    |
> | **Multi-task Learning**        | 15h 53m    |
>
> **Q2**
>
> Thanks for the suggestion. We additionally conduct experiments on out-of-domain generalization. Specifically, in the ViT-B/32 merging setup, we apply the meta-learning procedure and merge a unified model from 6 downstream tasks. We then perform subsequent adaptation and evaluation on 2 unseen tasks to simulate out-of-domain scenarios (follow the setting in [3]). We compare our method against several baselines, including Task Arithmetic[1], Ties-Merging[2], AdaMerging[3], and AdaMerging++[4]. We evaluate under two configurations, where the unseen tasks are MNIST & EuroSAT in the first and RESISC45 & SVHN in the second. The results are summarized below.
>
> | Method                        | SUN397 | Cars  | RESISC45 | DTD   | SVHN  | GTSRB | Avg. | *UNSEEN:* | MNIST | EuroSAT | Avg. |
> |-------------------------------|--------|-------|----------|-------|-------|-------|---------|-------|-------|---------|---------|
> | Task Arithmetic[1]  | 63.3   | 62.4  | 75.1     | 57.8  | 84.6  | 80.4  | 70.6    | | 77.2  | 46.2    | 61.7    |
> | Ties-Merging[2]     | 67.8   | 66.2  | 77.2     | 56.7  | 77.1  | 70.9  | 69.3    | | 75.9  | 43.3    | 59.6    |
> | AdaMerging[3]             | 65.2   | 65.9  | 88.5     | 61.1  | 92.2  | 91.5  | 77.4   | | 84.0  | 56.1    | 70.0    |
> | AdaMerging++[3]           | 68.2   | 67.6  | 86.3     | 63.6  | 92.6  | 89.8  | 78.0   | | 83.9  | 53.5    | 68.7    |
> | **MetaMerging (ours)** | 73.9 | 69.3 | 94.2 | 69.0 | 97.4 | 98.7 | 83.7 | | 99.5 | 94.4 | 96.9 |
>
> | Method                        | SUN397 | Cars  | GTSRB  | EuroSAT | DTD   | MNIST | Avg. | *UNSEEN:* | RESISC45 | SVHN  | Avg. |
> |-------------------------------|--------|-------|--------|---------|-------|-------|-----|----|----------|-------|---------|
> | Task Arithmetic[1] | 64.0   | 64.0  | 75.2   | 87.7    | 57.0  | 95.7  | 73.9   | | 52.3     | 44.9  | 51.1    |
> | Ties-Merging[2]     | 68.0   | 67.1  | 67.7   | 78.4    | 56.5  | 92.8  | 71.8  | | 58.7     | 49.2  | 53.9    |
> | AdaMerging[3]            | 67.1   | 67.8  | 94.8   | 94.4    | 59.6  | 98.2  | 80.3  |  | 50.2     | 60.9  | 55.5    |
> | AdaMerging++[3]          | 68.9   | 69.6  | 91.6   | 94.3    | 61.9  | 98.7  | 80.8  |  | 52.0     | 64.9  | 58.5    |
> | **MetaMerging (ours)** | 73.2 | 72.4 | 98.3 | 99.6 | 65.1 | 99.6 | 84.7  |  | 88.7 | 96.6 |  92.6  |
>
> The results show that MetaMerging achieves the best performance in both two configurations, outperforming the strongest baseline by a clear margin on seen and unseen tasks. This indicates that our meta-learned coefficients generalize beyond the training domains and remain stable under distribution shift.
>
>
>
> [1]ICLR'2023 Editing models with task arithmetic. https://openreview.net/forum?id=6t0Kwf8-jrj
> [2]NeurIPS'2023 TIES-Merging: Resolving Interference When Merging Models. https://openreview.net/forum?id=xtaX3WyCj1
> [3]ICLR'2024 AdaMerging: Adaptive Model Merging for Multi-Task Learning. https://openreview.net/forum?id=nZP6NgD3QY

---

### Official Review · Reviewer_oWra · 2025-10-31

**Soundness:** 3
**Presentation:** 2
**Contribution:** 2
**Rating:** 4
**Confidence:** 5

**Summary:**

This paper proposes MetaMerging, a meta-learning framework for model merging under the pretrain-finetune paradigm. Instead of treating merging coefficients as fixed or manually tuned hyperparameters, the method meta-learns the coefficients so that the unified merged model yields better downstream task-specific adapter training. Experiments on vision and NLP tasks show improved unified and adapter-augmented performance over prior merging methods.

**Strengths:**

1, Motivated formulation. \
Clearly identifies the overlooked importance of merging coefficients in task-vector–based model merging and motivates learning them dynamically.

2, Method soundness. \
Uses a meta-learning framework inspired by MAML to optimize coefficients for better downstream adaptation, which is conceptually meaningful and aligned with the merging problem structure.

**Weaknesses:**

1, Adapter-dependent benefit: \
Improvements are strongest when adapters are added post-merge, raising questions on the intrinsic generalization of the unified model alone vs. the joint effect with adapters.

2, Limited analysis on when meta-merging helps: \
There is insufficient theoretical or empirical characterization of:

2.1 when adaptive coefficients matter most,

2.1 how task similarity affects meta-learning benefit,

2.3 failure cases (e.g., conflicting tasks, negative transfer).

3, Limited novelty vs. MAML:

The contribution mainly adapts MAML to the setting of merging coefficients. The algorithmic innovation is relatively incremental; most complexity lies in applying known meta-learning ideas to merging.

4, Limited evaluation:

There should be some experiments on merging LLMs.

5. Limited baselines:

There should be comprasion with lossless methods like: EMR-Merging, Talls-Mask and Free-Merging.

6, The method still rely on data samples, which is not data-free and may face more challenges for LLMs.

**Questions:**

1，Adapter structure comparability & design choices:

The performance gains appear closely tied to the adapter modules. Could the authors clarify: What specific adapter architecture is used (e.g., LoRA, bottleneck adapters, MoE heads)? Are adapter capacities kept strictly equal across baselines to ensure a fair comparison?
Have you tested whether the method still holds with different adapter forms (e.g., shared vs. per-task adapters, low-rank variants, different insertion layers)? Since adapters play a key role in the pipeline, a more detailed justification and ablation of adapter design would strengthen the claim that improvements primarily come from better merging rather than from architectural choices.

2，Compared with simple coefficient search:

What is the actual meta-training overhead compared to simple coefficient search (e.g., CMA-ES or Bayesian tuning)? Is the method scalable to very large models or many tasks?

3, How many data samples you used for feature loss computing, if the loss can be replaced by L1 or others ?

---

> ### Author Response · Authors · 2025-11-26
> **Response to Weakness 1 & 2**
>
> Thank you for the thoughtful feedback. We provide our detailed response as follows:
>
> **W1**
>
> Our method benefits from the interaction between the unified model and the post-merge adapters, rather than from the adapters alone. While the unified model by itself sometimes brings only moderate gains compared to other merging strategies, it is explicitly optimized to serve as a good initialization for subsequent adapter training, not to maximize standalone accuracy. The meta-learned coefficients encourage more consistent task-shared representations, which makes the unified model particularly amenable to fast and effective adaptation. As a result, the same lightweight adapters can extract more task-specific knowledge when built on top of our unified model, leading to a much larger overall performance improvement than what is achievable by either component alone.
>
> This effect is quantitatively supported by the ablations in Table 5 and Table 6. Under the same adapter architecture and training protocol, attaching adapters to the unified model produced by Weight Averaging (`Weight Averaging w/ adapters` in the tables) yields average accuracies of 80.0% (ViT-B/32) and 85.9% (ViT-L/14), whereas MetaMerging reaches 87.2% and 93.4%, respectively. At the same time, the MetaMerging unified model without adapters remains competitive with, or comparable to, other merging baselines, showing that our method does not sacrifice the intrinsic quality of the merged backbone. Instead, meta-learning reshapes the unified model to be more "adapter-friendly": as shown by the faster loss decrease and lower final alignment loss in our adapter-training curves, the meta-learned unified model provides a better starting point for adaptation. Overall, these results indicate that the performance gains are not merely adapter-dependent, but arise from the synergistic combination of a meta-learned unified model and task-specific adapters.
>
>
> **W2**
>
>
> We appreciate the reviewer's insightful comment regarding understanding when meta-merging provides the most benefit. Based on theoretical considerations and our current experimental results, we have the following observations:
>
> 2.1 When adaptive coefficients matter most.
>
> Our primary goal is to learn coefficients that adapt to the heterogeneity across tasks and models, rather than to simply outperform uniform averaging in every scenario. Empirically, we observe that the benefit of meta-learned coefficients is most pronounced when task heterogeneity is moderate to high, since tasks with similar representations require less adjustment, while highly divergent tasks benefit from learned weighting. In more homogeneous settings, where all models are similar and trained on closely related tasks, the gap between learned and fixed coefficients naturally shrinks, and meta-merging behaves close to a robust version of weighted averaging.
>
> 2.2 Effect of task similarity.
>
> While a full theoretical characterization of task similarity is beyond the scope of this work, our benchmarks already span a spectrum from related tasks to clearly cross-domain ones. Specifically, on the vision side we merge eight ViT-B/32 and ViT-L/14 models fine-tuned on SUN397, Cars, RESISC45, EuroSAT, SVHN, GTSRB, MNIST, and DTD, which cover scenes, objects, remote sensing, digits, traffic signs, and textures. As shown in Table 1 and the ablation in Table 5, MetaMerging achieve higher average accuracy of the merged ViT-B/32 model than using unmerged pre-training models, with especially large gains on more `outlier` tasks such as EuroSAT and MNIST. On the NLP tasks, we merge GPT-2 models fine-tuned on seven GLUE tasks that exhibit diverse problem types: linguistic acceptability, sentiment, paraphrase, similarity, and several forms of entailment and NLI, as summarized in Table 11. In this setting, MetaMerging raises the average accuracy with the largest per-task improvements occurring on structurally distinct, low-resource tasks such as CoLA, MRPC, and RTE. Intuitively, in these heterogeneous tasks, the meta-learner can exploit complementary inductive biases and learn coefficients that emphasize the most relevant experts per task, whereas in more homogeneous tasks, averaging coefficients are already closer to optimal.
>
>
> 2.3 Failure cases and negative transfer.
>
> We acknowledge that meta-merging is not guaranteed to help for every task, especially in the presence of strongly conflicting objectives. In our current experiments, we already observe a few tasks where the improvement is small or where meta-merging is on par with the best single model, which can be interpreted as mild negative transfer or task conflict. We will explicitly point out these cases and discuss them as limitations: when a task is an outlier with very different input statistics, the optimal strategy may be to down-weight some experts or even exclude them via gating, which we view as a promising direction for future work.

---

> ### Author Response · Authors · 2025-11-26
> **Response to Weakness 3 & 4**
>
> **W3**
>
>
> We appreciate the reviewer's observation. While inspired by MAML, our goal is not to propose a completely unrelated optimization scheme, but to show that meta-learning is an effective and non-trivial way to solve the specific problem of model merging. In this sense, the novelty of our work lies less in inventing a new meta-learning method and more in how we reformulate and extend the merging problem so that meta-learning over merging coefficients becomes both effective and practical.
>
> Technically, there are several key differences from standard MAML beyond simply `plugging MAML into merging`. Classic MAML learns a shared initialization of network parameters that is subsequently adapted via gradient descent on each task. In contrast, we (1) keep all expert models frozen, (2) meta-learn structured, layer-wise merging coefficients over this pool of experts, and (3) combine this with lightweight, post-merge adapters. This leads to a different bi-level problem: the inner loop adapts only the adapters on top of a merged backbone defined by the current coefficients, and the outer loop updates these coefficients under simplex and consistency constraints to optimize post-adaptation performance. Designing this formulation required nontrivial modifications such as how coefficients are parameterized and regularized and how gradients are propagated through merging and adapter training.
>
> Thus, although influenced by meta-learning, the algorithm is not a direct adaptation but a new formulation tailored for model merging, as supported by experimental gains.
>
>
>
> **W4**
>
> Thanks for raising this concern. Due to current computational and GPU memory constraints as well as limited time, we are unable to run experiments on large language models at this stage. Nevertheless, we conduct extensive experiments on ViT and GPT-2, which serve as standard benchmarks in the majority of the model merging literature [1][2][3][4][5][6]. These benchmarks are representative and sufficient to demonstrate the effectiveness and generality of our method, and we expect the observed trends to largely carry over to LLMs once resources allow.
>
>
> [1]NeurIPS'2024 EMR-Merging: Tuning-Free High-Performance Model Merging. https://openreview.net/forum?id=lYdjzx3DYu
> [2]ICCV'2024 FREE-Merging: Fourier Transform for Efficient Model Merging. https://arxiv.org/pdf/2411.16815
> [3]ICML'2024 Localizing Task Information for Improved Model Merging and Compression. https://openreview.net/forum?id=DWT9uiGjxT
> [4]ICLR'2024 AdaMerging: Adaptive Model Merging for Multi-Task Learning. https://openreview.net/forum?id=nZP6NgD3QY
> [5]ICML'2024 Representation Surgery for Multi-Task Model Merging. https://openreview.net/forum?id=Sbl2keQEML
> [6]fusionbench FusionBench: A Comprehensive Benchmark of Deep Model Fusion. https://openreview.net/forum?id=a0sK0foX3p

---

> ### Author Response · Authors · 2025-11-26
> **Response to Weakness 5 & 6**
>
> **W5**
>
> Here we present the comparison with three new baseline methods: EMR-Merging [1], FR-Merging [2], and FREE-Merging(FR-Merging with experts) [2] (when merging 8 ViT-B/32 models).
>
> | Method               | SUN397 | Cars | RESISC45 | EuroSAT | SVHN | GTSRB | MNIST | DTD  | Avg. |
> |----------------------|--------|------|----------|---------|------|-------|-------|------|------|
> | **MetaMerging (ours)** | 74.2   | 71.9 | 92.0     | 99.4    | 97.1 | 98.1  | 99.6  | 64.9 | 87.2 |
> | EMR-Merging [1]      | 75.2   | 72.8 | 93.5     | 99.5    | 96.9 | 98.1  | 99.6  | 74.4 | 88.7 |
> | FR-Merging [2]       | 66.2   | 64.5 | 77.2     | 90.1    | 85.4 | 82.3  | 98.5  | 60.0 | 78.1 |
> | FREE-Merging [2]     | 77.1   | 78.2 | 93.4     | 99.5    | 96.3 | 98.2  | 99.5  | 75.4 | 89.7 |
>
>
> Although our method does not surpass EMR-Merging and FREE-Merging in raw accuracy, several key aspects should be noted:
> (1) Mask-based merging methods (e.g., EMR-Merging [1], Talls-Mask [3]) require maintaining a separate model with a task-specific mask at inference time, which disables parallel inference across tasks and prevents a single unified deployment. This issue was explicitly raised by reviewers of EMR-Merging; see Reviewer S3RQ in [1].
> (2) In paper [2], FR-Merging is a training-free baseline, whereas FREE-Merging is a significantly more complex trainable method. FREE-Merging further introduces an MoE architecture where each expert is derived from task vectors and requires training a parameterized router (see Table 13 in [2]). These additional experts lead to substantial storage overhead and higher inference-time computational cost.
>
> Compared with the above, our method remains parameter-efficient, keeps both training time and inference cost manageable, and supports parallel inference for all tasks using a single shared backbone. In the model merging literature, higher accuracy often comes with substantially higher training or parameter costs. In contrast, our work aims to strike a balanced trade-off between performance, efficiency, and deployability, which we believe is essential for scalable multi-task deployment.
>
> [1]NeurIPS'2024 EMR-Merging: Tuning-Free High-Performance Model Merging. https://openreview.net/forum?id=lYdjzx3DYu
> [2]ICCV'2024 FREE-Merging: Fourier Transform for Efficient Model Merging. https://arxiv.org/pdf/2411.16815
> [3]ICML'2024 Localizing Task Information for Improved Model Merging and Compression. https://openreview.net/forum?id=DWT9uiGjxT
>
>
>
>
> **W6**
>
> We would like to clarify that although our training is not data-free, it is entirely label-free throughout. Our framework only needs raw text to define task-specific inputs, which makes data collection considerably easier for LLMs.
> This also suggests that synthetic text produced by an LLM could, in principle, be useful for our meta-training tasks, since the method does not rely on annotated supervision.

---

> ### Author Response · Authors · 2025-11-26
> **Response to Question 1 & 2 & 3**
>
> **Q1**
>
> Without loss of generality, we use a uniform bottleneck adapter (dimension = 64) across all baselines, inserted after the Transformer feedforward blocks. We have clarified this design choice in Appendix C.
>
> To test the generality of our method and rule out architectural bias, we additionally performed experiments using LoRA adapters (rank = 8, chosen to keep the number of trainable parameters comparable) while keeping all other training settings unchanged. The results are shown below:
>
> | Method            | SUN397 | Cars | RESISC45 | EuroSAT | SVHN | GTSRB | MNIST | DTD  | Avg.  |
> |-------------------|--------|------|----------|---------|------|-------|-------|------|------|
> | **LoRA adapters**          | 73.0   | 74.3 | 90.1     | 98.5    | 96.2 | 98.9  | 99.4  | 66.4 | 87.1 |
> | **bottleneck adapters**| 74.2   | 71.9 | 92.0     | 99.4    | 97.1 | 98.1  | 99.6  | 64.9 | 87.2 |
>
> We observe that the performance difference between LoRA adapters and bottleneck adapters is small. This confirms that the improvement of our method is not due to a specific adapter architecture, but rather stems from the unified model and the subsequent adaptation mechanism. We will include this new comparison table in the revised paper.
>
>
> **Q2**
>
> In practice, search-based methods such as CMA-ES or Bayesian tuning are substantially more expensive than our meta-learning procedure. CMA-ES typically requires a population of 32–64 candidates per iteration and tens of iterations, while Bayesian tuning usually needs dozens of function evaluations to obtain a reliable surrogate model. Since the coefficients must be chosen before adapter training, each evaluation triggers a full adapter-training run over all tasks, making the total cost easily one to two orders of magnitude larger than a single adapter-training process.
>
> In contrast, in our ViT-B/32 merging experiment, the adapter training takes 46 minutes, and our entire meta-learning procedure finishes in 91 minutes, showing that our method is significantly more efficient than coefficient-searching approaches.
>
> Additionally, the computational cost of meta-learning mainly depends on the adapter parameter size and the number of inner-loop steps, making it scalable to very large backbone models and to a larger number of tasks by appropriately adjusting the learning setup.
>
> **Q3**
>
> During the training process, we sample a batch of inputs from each dataset whenever data sampling is required (lines 11 and 20 in Algorithm 1). In the code, we control this process using two hyperparameters: `meta_batch_size` and `meta_batch_size_test`. Specifically, in the inner loop, we sample 8 input instances per dataset for adapter updates, while in the outer loop, we sample 4 input instances per dataset for meta-testing to update the merging coefficients (ViT-B/32 merging experiment). The entire meta-learning procedure is trained for 3000 epoch iterations.
>
> During the design of our method, we conducted experiments to evaluate various loss functions and found that L2 loss consistently yields the best performance, which is why it was adopted. Here, we present a performance comparison between L2 loss and L1 loss.
>
> | Method               | SUN397 | Cars | RESISC45 | EuroSAT | SVHN | GTSRB | MNIST | DTD  | Avg. |
> |----------------------|--------|------|----------|---------|------|-------|-------|------|------|
> | **L2 loss** | 74.2   | 71.9 | 92.0     | 99.4    | 97.1 | 98.1  | 99.6  | 64.9 | 87.2 |
> | **L1 loss** | 75.6 | 71.2 | 94.4 | 99.0 | 97.2 | 95.5 | 99.4 | 61.7 | 86.7 |

---

### Author Response · Authors · 2025-12-03
**Rebuttal Summary (2/2)**

**Reviewer 5icd (Score: 6)**
This reviewer finds the work meaningful and well-structured but requests clearer differentiation from adaptive weighting methods and further discussion on scalability.

- Main concerns:
(1) Need clearer differentiation from existing adaptive merging methods.
(2) Benchmarks may be limited; requested more diverse domains and real-world settings.
(3) High-level accessibility of Section 3 should be improved.

- Response:
(1) We clarified that MetaMerging differs fundamentally from prior adaptive weighting by meta-learning coefficients optimized for post-adaptation performance via bi-level optimization.
(2) We cited extensive prior literature to justify that the benchmarks we used are sufficiently diverse and representative of heterogeneous real-world scenarios.
(3) We added a high-level overview before Section 3 to improve clarity.


**Reviewer 9SxD (Score: 6)**
This reviewer appreciates the overall contribution but asks for additional modern baselines and clarification on scalability, memory cost, and training time.

- Main concerns:
(1) Missing newer baselines (MoE-based and gradient-space merging)
(2) Concerns about meta-learning scalability to large models.
(3) Memory and runtime higher than Surgery[4]; improvement magnitude.

- Response:
(1) We added comparisons with WEMoE[5] (MoE-based), DOGE[6], and SuperMerge[7] (gradient-space).
(2) We clarified that our method uses first-order meta-learning and only updates few parameters, making it feasible for LLMs.
(3) We provided memory usage and training-time comparisons, showing the cost remains moderate and significantly lower than multi-task training.



In the newly submitted manuscript, we have added two parts:
- High-level description of the motivation, purpose, and methodology at the beginning of Section 3
- Comparison of the training time of existing model merging methods in Table 4.

The revised text and newly added sections are highlighted in blue.

We hope that this summary helps reduce the AC's workload and facilitates a more efficient evaluation of our submission and rebuttal. We sincerely thank the ACs, SAC, PCs, and all reviewers for their time, effort, and dedication in handling the challenges caused by the recent incident.

Best regards,
The Authors

[1]NeurIPS'2024 EMR-Merging: Tuning-Free High-Performance Model Merging. https://openreview.net/forum?id=lYdjzx3DYu
[2]ICCV'2024 FREE-Merging: Fourier Transform for Efficient Model Merging. https://arxiv.org/pdf/2411.16815
[3]ICML'2025 Whoever Started the interference Should End It: Guiding Data-Free Model Merging via Task Vectors. https://openreview.net/forum?id=xR9msNaREW
[4]ICML'2024 Representation Surgery for Multi-Task Model Merging. https://openreview.net/forum?id=Sbl2keQEML
[5]ICML'24 Merging Multi-Task Models via Weight-Ensembling Mixture of Experts. https://openreview.net/forum?id=nLRKnO74RB
[6]ICML'25 Modeling Multi-Task Model Merging as Adaptive Projective Gradient Descent. https://openreview.net/forum?id=EqoKRSR5Pa
[7]ICML'25 SUPERMERGE: An Approach For Gradient-Based Model Merging. https://openreview.net/forum?id=lIdc5DUplq

---

### Author Response · Authors · 2025-12-03
**Rebuttal Summary (1/2)**

Dear AC, SAC, PCs, and Reviewers,

We are aware that the recent incident has impacted the ICLR community, and we sincerely regret the situation. We thank the PCs for the timely actions and the new protocol, and we appreciate the newly assigned AC for taking on the additional workload. We also thank all reviewers for their valuable and constructive feedback, even though the incident prevented the continuation of the rebuttal discussion. We understand that the newly assigned AC needs to evaluate our response and make the final decision, and we sincerely appreciate the efforts involved.

To assist the AC in this process, we provide a concise summary of the main reviewer concerns and our corresponding responses, as well as a brief description of the situation surrounding this submission.

### **Rebuttal Summary:**

Brief description of the situation: All reviewers recognize the contributions and strengths of our work. We have provided detailed responses to every concern raised and believe that most of them have been adequately addressed. Unfortunately, the incident occurred shortly after we submitted the rebuttal, leaving us no opportunity to receive additional feedback or continue the discussion. Thus, the current status for all reviewers remains "no response".

Below is a summary of the major concerns and our responses for all reviewers.

**Reviewer oWra (Score: 4)**
This reviewer mainly raised questions about our empirical analysis while acknowledging the soundness and rationale of the proposed method, and we addressed all concerns in the rebuttal.

- Main concerns:
(1) Uncertainty about whether improvements primarily come from adapters.
(2) Limited analysis of when meta-merging helps and potential failure cases.
(3) Missing comparison with several baselines.
(4) Choices of loss functions and adapter designs.

- Response:
(1) We clarified that the gains arise from the synergy between the meta-learned unified model and the adapters, supported by ablations in Tables 5 and 6.
(2) We provided detailed analysis of task similarity and heterogeneity, and we reported mild failure cases.
(3) We added comparisons with EMR-Merging[1], FR-Merging[2] and FREE-Merging[2], along with a discussion of their respective strengths and limitations, such as computational cost and memory usage.
(4) We added experiments reporting comparisons of loss-function choices (with L1 loss) and comparisons of adapter designs (with LoRA).


**Reviewer uyTM (Score: 2)**
This reviewer focused mainly on experimental weaknesses, and we provided extensive new experiments and clarifications in the rebuttal.

- Main concerns:
(1) Missing comparison with WUDI-Merging[3] and other state-of-the-art merging baselines.
(2) The GPT-2 baseline was considered weak, and recent llama-based LMs were not evaluated.
(3) Requests for out-of-domain generalization experiments.

- Response:
(1) We added a full comparison with WUDI-Merging[3]. We also clarified that many recent methods require auxiliary modules, leading to significantly higher memory usage or training burden, whereas our method strikes a better balance between performance and efficiency.
(2) We explained the rationale for using GPT-2, which aligns with most existing model-merging works. llama-based experiments are beyond the current scope and left as future work.
(3) We conducted new out-of-domain generalization experiments (merging 6 tasks and evaluating on 2 unseen tasks), where MetaMerging achieved the best performance on both seen and unseen tasks.

---

### Meta-Review · Area_Chair_9xnj · 2026-01-07

**Summary:**

This paper introduces MetaMerging, a meta-learning-based framework for adaptive model merging in multi-task learning. The proposed method leverages meta-learning to automatically optimize merging coefficients, which results in improving the unified model's generalization and facilitating efficient training of task-specific adapters. The AC has carefully reviewed the paper independently and summarized the reviewers’ main concerns and questions as follows:

* Reviewer oWra: adapter-dependent benefit (improvements are significant when adapters are added post-merge, bringing up questions about the intrinsic generalization of the unified model alone vs. with adapters); limited analysis of when meta-merging is beneficial (with several detailed questions); limited novelty relative to MAML; limited evaluation and baselines; questionable applicability to large language models; and unclear impact of meta-training compared to simple coefficient search.

* Reviewer uyTM: missing SOTA comparisons; marginal performance improvements over existing methods; and NLP baselines based on relatively small models that do not reach LLM-scale.

* Reviewer 5icd: limited novelty; experiments only on benchmark-level without large-scale or real-world multi-task scenarios; concerns about how the proposed meta-learning differs from prior adaptive weighting methods with marginal performance improvements.

* Reviewer 9SxD: missing or limited comparisons; marginal improvements; inefficient memory requirements; and limited applicability to large language models.

**Reviewer Concerns:**

The shared critical concerns can be classified into four main categories: (1) limited comparisons, (2) marginal performance improvements, (3) unclear distinctions between the proposed meta-learning approach and existing adaptive methods, such as adaptive weighting or simple coefficient search, and (4) limited experimental scale, including the applicability to LLMs.

The AC believes that model-merging methods for multi-task learning are not necessarily applicable to LLMs; while such an extension would certainly be valuable, applicability to LLMs is beyond the intended scope of this line of work. However, concerns remain about limited comparisons, particularly the absence of certain SOTA baselines (e.g., Twin-Merging and WeMoE). While the rebuttal includes comparisons with WeMoE, Twin-Merging is still missing. Moreover, although not explicitly raised by the reviewers, there are additional recent SOTA methods that were not considered (e.g., ProDistill, ProbSurgery); while it may not be necessary to address all of them at this stage, the AC recommends considering them in future revisions. Furthermore, the reported performance gains appear marginal; the authors made an effort to address reviewers' requests by incorporating additional literature and comparisons. Nevertheless, the performance gains remain modest. While an average improvement of +1.1% over Surgery may not be marginal in isolation, the relative advantage becomes less compelling when additional comparisons from the expanded literature are incorporated. As such, it is unlikely that this issue can be fully resolved within the current review cycle.

Furthermore, the AC shares some concerns raised by the reviewers that do not appear to be fully addressed: (1) it remains unclear why a meta-learning formulation is necessary and what concrete benefits it provides over simpler adaptive methods and some external adapter-like modules; and (2) it is questionable that the unified model itself benefits from the merging effect (e.g., improved generalization) without the use of adapters (as Table 5 indicates that the adapter-free setting underperforms).

Finally, the AC encourages the authors to address the above issues in a future submission and further recommends conducting more comprehensive experiments, such as evaluating on larger task suites (e.g., 14 or 20 tasks rather than 8 tasks), and ensuring coverage of recent baselines for comparison.

**Reviewer Scores:**

Unfortunately, no reviewers participated in the discussion phase, which was not ideal for the authors. However, it appears that significant concerns overlapped, which may have led to limited further discussion. It is unlikely that the scores will change, even if further discussion continues.

---

### Decision · Program_Chairs · 2026-01-26

Reject